# FSD-CAP: Fractional Subgraph Diffusion with Class-Aware Propagation for Graph Feature Imputation

**Xin Qiao**[1]*, **Shijie Sun**[1]*, **Anqi Dong**[2]*, **Cong Hua**[1], **Xia Zhao**[1], **Longfei Zhang**[3], **Guangming Zhu**[1], **Liang Zhang**[1]†

[1]Xidian University, Xi'an, China
[2]KTH Royal Institute of Technology, Stockholm, Sweden
[3]Hunan Institute of Advanced Technology, Hunan, China
`{xinqiao,ssj,chua}@stu.xidian.edu.cn, anqidong.andy@gmail.com`
`{zhaoxia,gmzhu,liangzhang}@xidian.edu.cn`

## Abstract

Imputing missing node features in graphs is challenging, particularly under high missing rates. Existing methods based on latent representations or global diffusion often fail to produce reliable estimates, and may propagate errors across the graph. We propose FSD-CAP, a two-stage framework designed to improve imputation quality under extreme sparsity. In the first stage, a graph-distance-guided subgraph expansion localizes the diffusion process. A fractional diffusion operator adjusts propagation sharpness based on local structure. In the second stage, imputed features are refined using class-aware propagation, which incorporates pseudo-labels and neighborhood entropy to promote consistency. We evaluated FSD-CAP on multiple datasets. With $99.5\%$ of features missing across five benchmark datasets, FSD-CAP achieves average accuracies of $80.06\%$ (structural) and $81.01\%$ (uniform) in node classification, close to the $81.31\%$ achieved by a standard GCN with full features. For link prediction under the same setting, it reaches AUC scores of $91.65\%$ (structural) and $92.41\%$ (uniform), compared to $95.06\%$ for the fully observed case. Furthermore, FSD-CAP demonstrates superior performance on both large-scale and heterophily datasets when compared to other models. Codes are available at https://github.com/ssjcode/FSD-CAP.

## 1 Introduction

Graph Neural Networks (GNNs) are widely used for learning from graph-structured data, with successful applications in social networks (Bian et al., 2020), biology (Li et al., 2022), and recommendation systems (He et al., 2020). GNN architectures(Chen et al., 2023; Chien et al., 2020) always assume nodal features are fully observed, allowing information to be aggregated effectively from neighboring nodes. In practice, this assumption often fails. Node attributes are frequently missing due to privacy constraints, sensor failures, or incomplete data collection. High missing rates disrupt the message-passing process and significantly degrade model performance.

A variety of methods have been proposed for imputing missing features, including statistical estimators (Srebro et al., 2004), machine learning models (Chen & Guestrin, 2016), and generative approaches (Vincent et al., 2008). Recent work has shifted toward deep learning techniques that model the distribution of node attributes. These include latent space models that align observed features with learned embeddings (Chen et al., 2020; Yoo et al., 2022), and GNN-based architectures designed to operate on incomplete inputs (Taguchi et al., 2021). These approaches, which rely on correlations in both feature and graph structure, are effective under moderate missing rates but experience significant performance degradation as sparsity increases, ultimately falling below simple baselines like zero-filling or mean imputation in highly incomplete settings(You et al., 2020).

---

*Equal contribution.
†Corresponding author.

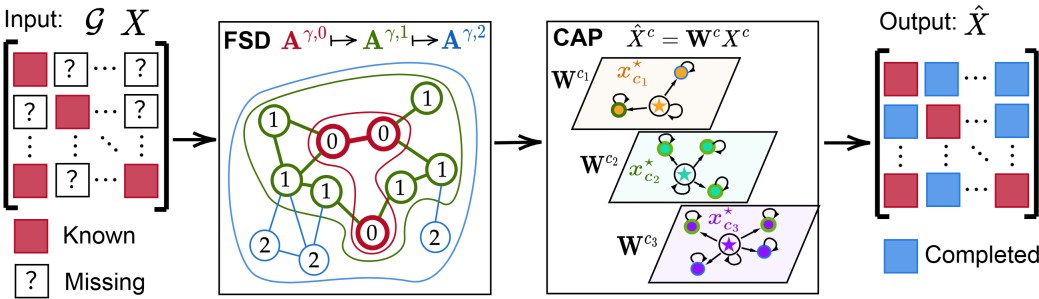

Figure 1: FSD-CAP Pipeline: Given graph $\mathcal{G}$ and partially observed feature matrix $X$, FSD-CAP recovers the full matrix $\hat{X}$. **(i) FSD:** Starting from observed nodes, it gradually expands the radius of the subgraph and performs progressive subgraph diffusion using fractional diffusion operators $\mathbf{A}^{\gamma,0}$, $\mathbf{A}^{\gamma,1}$, $\mathbf{A}^{\gamma,2}$, producing a preliminary imputed feature matrix. **(ii) CAP:** Based on the FSD-imputed features, pseudo-labels are assigned by a classifier to form class-wise graphs, each associated with class-specific $X^c$ and $\mathbf{W}^c$. Feature propagation within each class graph yields the final output $\hat{X}$.

An alternative class of methods, based on diffusion, propagates observed features across the graph under the assumption of node homophily (Rossi et al., 2022; Um et al., 2023; Wang et al., 2024). These methods are typically lightweight, parameter-free, and more robust under high missing rates. However, most diffusion approaches apply uniform propagation across all nodes, without accounting for local structure or propagation order. As a result, nearby reliable signals may be underused, particularly in sparse or large-scale graphs. Additionally, these methods often ignore variation in connectivity and feature distribution, leading to over-smoothing and reduced discriminative power in the imputed features.

We propose Fractional Subgraph Diffusion with Class-Aware Propagation (FSD-CAP), a diffusion-based framework for imputing missing features on graphs. The method is designed to improve robustness under feature sparsity and adapt to local structural variation. FSD-CAP consists of three components. First, a fractional diffusion operator modulates the sharpness of propagation based on local graph structure. This operator generalizes standard normalization by interpolating between uniform averaging and dominant-neighbor selection, allowing it to adapt to varied connectivity. Second, to reduce error accumulation from global diffusion, we introduce a graph-distance-guided subgraph expansion strategy. This mechanism begins with observed nodes and progressively includes less certain regions, enabling early reliable estimates and improved stability. Third, a class-aware refinement step uses pseudo-labels and neighborhood entropy to enhance the imputed features, promoting intra-class consistency and inter-class separation. In the tasks of semi-supervised node classification and link prediction on five benchmark datasets, this method consistently outperforms state-of-the-art imputation approaches and maintains robustness across a wide range of missing rates. Notably, on datasets such as CiteSeer and PubMed, FSD-CAP exhibits higher performance under extreme missing conditions compared to when utilizing fully observed features, thereby demonstrating its effectiveness in sparse scenarios. Furthermore, when compared to other models, FSD-CAP also achieves superior performance on two large-scale datasets and four heterophilous datasets, showcasing its strong adaptability (Appendix A.2.7 and Appendix A.2.8).

## 2 PROPOSED METHOD

### 2.1 PRELIMINARIES AND OVERVIEW

Our objective is to learn from graphs with incomplete node features, focusing on two common settings: *structural missing*, where some nodes have no observed features, and *uniform missing*, where entries are randomly missing across the feature matrix. The goal is to impute missing attributes in a way that enables robust representation learning, especially under high missing rates where standard GNNs fail. To this end, we propose a two-stage imputation framework designed to stabilize learning and adapt to local graph structure. The overall architecture is shown in Figure 1 and consists of the following three key components.

**(i) Fractional diffusion operator.** We generalize the standard diffusion matrix (Gasteiger et al., 2019) by introducing a fractional exponent that controls the sharpness of propagation (Section 2.2). This operator interpolates between uniform averaging and nearest-neighbor routing, allowing information to diffuse adaptively according to local graph structure and thus emphasizes *reliable signals* and mitigates *over-smoothing*.

**(ii) Progressive subgraph diffusion.** We propagate information through a structured, layer-wise expansion from observed to unobserved nodes (Section 2.3), instead of global diffusion (Rossi et al., 2022). At each step, only a localized subgraph is updated, thereby reducing error accumulation and improving stability. It prioritizes *easy-to-complete* nodes in early stages and gradually expands to uncertain regions.

**(iii) Class-level feature refinement.** We construct synthetic class-level features using pseudo-labels and neighborhood entropy (Section 2.4) to improve the discriminability of features. These signals are propagated within intra-class graphs to refine feature estimates and improve semantic consistency. This step promotes *intra-class coherence* and preserves *inter-class distinctiveness*, reducing over-smoothing across class boundaries.

Before presenting the technical details, we introduce the notation used throughout this section. Let $\mathcal{G} = (\mathcal{V}, \mathcal{E})$ denote an undirected graph, where $\mathcal{V} = \{v_1, v_2, \ldots, v_N\}$ is the set of $N$ nodes and $\mathcal{E} \subseteq \mathcal{V} \times \mathcal{V}$ is the set of edges. Each node $v_k$ is associated with a feature vector in $\mathbb{R}^F$, and we collect all node features in a matrix $X \in \mathbb{R}^{N \times F}$, where $X_{[k,:]}$ denotes the feature vector of node $v_k$. The topology of the graph is represented by an adjacency matrix $A \in \{0,1\}^{N \times N}$, where $A_{ij} = 1$ if $(v_i, v_j) \in \mathcal{E}$ and $A_{ij} = 0$ otherwise. The corresponding degree matrix $D \in \mathbb{R}^{N \times N}$ is diagonal, with entries $D_{ii} = \sum_{j=1}^{N} A_{ij}$. To represent missing features, we define a binary mask matrix $M \in \{0,1\}^{N \times F}$, where $M_{k\ell} = 1$ if the $\ell$-th feature of node $v_k$ is observed, and $M_{k\ell} = 0$ otherwise. Proofs of theoretical results in this section are included in the supplementary material.

## 2.2 FRACTIONAL DIFFUSION OPERATOR

Diffusion-based imputation methods typically rely on the symmetrically normalized adjacency matrix $\mathbf{A} = D^{-1/2}AD^{-1/2}$, which defines a lazy random walk over the graph (Rossi et al., 2022; Malitesta et al., 2024; Chen et al., 2016; Dong et al., 2025b;a). This operator assumes uniform mixing across neighbors, failing to account for differences in node distributions (Ji et al., 2023), which may result in over-smoothing and diminished feature discriminability. To address this, we introduce a fractional diffusion operator that adjusts the propagation behavior using a tunable sharpness parameter $\gamma > 0$. *The key idea is to amplify or suppress the relative influence of neighboring nodes through elementwise exponentiation followed by row normalization.* Specifically, the fractional diffusion matrix $\mathbf{A}^\gamma \in \mathbb{R}^{N \times N}$ is defined as

$$\mathbf{A}_{ij}^\gamma := (\mathbf{A}_{ij})^\gamma / \Big( \sum_{k=1}^{N} (\mathbf{A}_{ik})^\gamma \Big). \tag{1}$$

This transformation preserves the row-stochastic property of $\mathbf{A}$ while reweighting neighbor contributions based on edge strength. For $\gamma < 1$, weaker edges are amplified, resulting in smoother, more uniform propagation. For $\gamma > 1$, stronger edges are emphasized, leading to sharper, more localized diffusion. The standard normalized diffusion is recovered when $\gamma = 1$.

**Proposition 1 (Limiting behavior of $\mathbf{A}^\gamma$).** *Let $\mathbf{A}$ be the symmetric normalized adjacency matrix of a connected graph, and $\mathbf{A}^\gamma$ as in equation 1. We have*

$$\lim_{\gamma \to 0^+} \mathbf{A}_{ij}^\gamma = \begin{cases} \frac{1}{|\mathcal{N}(i)|} & \text{if } \mathbf{A}_{ij} > 0 \\ 0 & \text{otherwise} \end{cases} \quad \textit{and} \quad \lim_{\gamma \to \infty} \mathbf{A}_{ij}^\gamma = \begin{cases} 1 & \text{if } j \in \arg\max_k \mathbf{A}_{ik} \\ 0 & \text{otherwise} \end{cases}. \tag{2}$$

*where $\mathcal{N}(i) = \{j \mid \mathbf{A}_{ij} > 0\}$ denotes the set of neighbors of node $i$.*

Proposition 1 demonstrates that $\gamma$ acts as a locality parameter, i.e., small values encourage broad, uniform mixing across neighbors, while large values concentrate diffusion along the most prominent edge, resulting in highly localized propagation.

**Remark 1 (Super-diffusion and nearest-neighbor routing).** *For $\gamma > 1$, the diffusion process enters a* super-diffusion *regime, where high-weight edges exert disproportionately strong influence.*

*As $\gamma$ increases, the row weights in $\mathbf{A}^\gamma$ become increasingly concentrated around the largest entry, leading to highly localized propagation. In the limit $\gamma \to \infty$, the process reduces to deterministic routing, where each node transfers its mass entirely to its strongest neighbor.*

The effect of the fractional exponent $\gamma$ on the propagated features $X^\gamma := \mathbf{A}^\gamma X$, where $X$ is the input feature matrix, is formalized in Theorem 1.

**Theorem 1 (Fractional diffusion on feature propagation).** *Let $\mathbf{A}$ and $\mathbf{A}^\gamma$ denote the lazy transition and fractional diffusion operator as defined in equation 1, respectively. For any feature matrix $X \in \mathbb{R}^{N \times F}$, define the propagated features by $X^\gamma := \mathbf{A}^\gamma X$. Then for each node $i$, we have*

$$\lim_{\gamma \to 0^+} X^\gamma_{[i,:]} = \frac{1}{|\mathcal{N}(i)|} \sum_{j \in \mathcal{N}(i)} X_{[j,:]} \quad and \quad \lim_{\gamma \to \infty} X^\gamma_{[i,:]} = X_{[j^\star,:]},$$

*where $\mathcal{N}(i) = \{j \mid \mathbf{A}_{ij} > 0\}$ the neighborhood of $v_i$, and $j^\star \in \arg\max_k \mathbf{A}_{ik}$ is the index of its strongest neighbor.*

### 2.3 PROGRESSIVE SUBGRAPH DIFFUSION

In graph learning, nodes that are closer in the graph topology are generally considered to exhibit stronger feature similarity (You et al., 2019; Zhang & Chen, 2018). Motivated by this principle, we design a progressive subgraph diffusion strategy that replaces global propagation with a distance-aware, layer-wise process. Rather than diffusing information uniformly, the method propagates *features hierarchically from observed regions to unobserved ones.* Nodes closer to observed features are imputed more accurately (see, e.g., Appendix A.2.3), reinforcing the design choice of localized, incremental expansion to preserve reliable signals and limit early error propagation.

We now define subgraph construction at the level of individual feature dimensions. For each feature $\ell$, let $\mathcal{V}^\ell_+ := \{v_k \in \mathcal{V} \mid M_{k\ell} = 1\}$ denote the set of nodes with observed values, and let $\mathcal{V}^\ell_- := \mathcal{V} \backslash \mathcal{V}^\ell_+$ denote the set with missing values. The initial subgraph $\mathcal{G}^{(0)}$ is formed over $\mathcal{V}^\ell_+$ using adjacency relations inherited from the original graph $\mathcal{G}$. Since observed features may be spatially sparse or fragmented, $\mathcal{G}^{(0)}$ typically consists of multiple disconnected components. Next, to complete the feature progressively, we expand the subgraph in layers by incorporating nodes from $\mathcal{V}^\ell_-$ based on their shortest-path distance to $\mathcal{V}^\ell_+$. At layer $m$, the subgraph $\mathcal{G}^{(m)}$ includes all nodes within distance $m$ of $\mathcal{V}^\ell_+$, forming the radius-$m$ neighborhood $\mathcal{V}^{(m)}$, along with all edges among those nodes. As $m$ increases, the subgraph expands outward and disconnected regions gradually merge. In the limit, when $m$ equals the graph diameter, the subgraph $\mathcal{G}^{(m)}$ recovers the full graph $\mathcal{G}$.

At each layer-$m$ subgraph, we apply fractional diffusion described in Theorem 1 independently to each connected component. Let $\mathcal{G}^{(m)}_i$ denote the $i$-th connected component of $\mathcal{G}^{(m)}$, with corresponding adjacency matrix $A^{(m)}_i$. The fractional diffusion operator on this component is defined as $\mathbf{A}^{\gamma,m}_i := \mathbf{A}^\gamma(\mathcal{G}^{(m)}_i)$, using the formulation in equation 1. Let $x^{(m)}_i(t)$ denote the value of node $v_i$ in each feature channel at iteration $t$, the diffusion update is thus given by

$$x^{(m)}_i(t) = \sum_{v_j \in \mathcal{N}^{(m)}(i)} \mathbf{A}^{\gamma,m}_{ij} \cdot x^{(m)}_j(t-1), \tag{3}$$

where $\mathcal{N}^{(m)}(i)$ is the neighborhood of node $v_i$ in $\mathcal{G}^{(m)}$.

As the subgraph expands with increasing $m$, it gradually includes more distant nodes, which may carry unreliable or noisy information. This expansion can degrade the accuracy of features imputed in earlier layers. To address this, we introduce a *retention* mechanism that stabilizes updates by blending new estimates with those from previous layers. Additionally, we enforce a boundary condition to preserve observed features throughout the diffusion process. Let $M$ denote the binary mask matrix. At each iteration $t = 1, \ldots, K$, the channel-wise update rule for node $v_i$ is given by

$$x^{(m)}_i(t) = x^{(m)}_i(0) \odot M_i + \left( x^{(m)}_i(t) + \lambda x^{(m-1)}_i(K) \right) \odot (1 - M_i), \tag{4}$$

where $\odot$ denotes the Hadamard (element-wise) product and $x^{(m-1)}_i(K)$ represents the converged result of node $v_i$ from the previous layer after $K$ iterations that weighted by $\lambda$.

This update ensures that observed features remain fixed across all iterations, while missing values are updated based on a blend of current estimates and the previous layer's outputs. As $m$ increases, this retention mechanism promotes stability and gradual refinement, improving the reliability of imputed features across the diffusion process. After $K$ propagation steps using the update rules in equation 3 and equation 4, we obtain a refined estimate $x_u^{(m)}(K)$ for each node $u \in \mathcal{V}^{(m)}$ in each feature channel, corresponding to the $m$-th layer subgraph $\mathcal{G}^{(m)}$. As $K \to \infty$, the recursive updates converge to a fixed point. The following result establishes convergence of this process under fractional diffusion.

**Theorem 2 (Convergence of subgraph diffusion).** *Let $A^{(m)}$ be the adjacency matrix of the $m$-th layer subgraph $\mathcal{G}^{(m)}$, and let $\mathbf{A}^{\gamma,m}$ denote its fractional diffusion matrix as defined in equation 1. Let $x^{(m)} \in \mathbb{R}^{|\mathcal{V}^{(m)}|}$ be the feature vector in channel $\ell$, where missing entries are initialized to zero. Let $\lambda > 0$ denote the retention coefficient, and $M$ be the binary mask vector for observed entries in channel $\ell$. Define the update sequence by*

$$x^{(m)}(t) = x^{(m)}(0) \odot M + \left( \mathbf{A}^{\gamma,m} x^{(m)}(t-1) + \lambda x^{(m-1)}(K) \right) \odot (1-M), \quad t = 1, \ldots, K$$

*Then, for sufficiently large $K$, the sequence $x^{(m)}(t)$ converges to a fixed (unique) state.*

Theorem 2 guarantees that the iterative update process over each subgraph leads to a well-defined steady-state solution for missing features. By incorporating a retention factor, the method balances the influence of prior estimates and newly propagated information, thereby limiting error accumulation across layers. As the subgraph expands with increasing radius, progressively more nodes are included, and more structural context is captured. Theorem 3 shows that under mild assumptions, this layer-wise refinement converges to the solution that would have been obtained by applying the diffusion update over the entire graph at once. In this way, the global behavior of the model is recovered as the natural limit of consistent local operations.

**Theorem 3 (Global convergence via progressive subgraph expansion).** *Let $\mathcal{G}^{(m)}$ be the $m$-hop expansion of the observed node set $\mathcal{V}_+^\ell$ in channel $\ell$, and let $x^{(m)}$ be the corresponding feature estimate after applying the masked fractional diffusion update defined in Theorem 2. Assume the graph $\mathcal{G}$ is connected and the diffusion sharpness parameter $\gamma$ is finite. Then, as $m \to M_{\max}$ and $\mathcal{G}^{(m)} \to \mathcal{G}$, the final estimate $x^{(m)}(\infty)$ converges to the steady-state solution of the full-graph diffusion update.*

## 2.4 Class-level feature refinement

After computing the feature matrix using fractional and subgraph diffusion (Sections 2.2 and 2.3), we perform a class-guided refinement step. This stage is motivated by the observation that, under high missing rates, most features must be inferred from a sparse set of observed values. Diffusion alone tends to blur discriminative patterns, especially when semantic boundaries are inherently weak. Class-level propagation addresses this issue by injecting semantic structure into the imputation process to improve feature quality under severe sparsity.

We begin by assigning pseudo-labels to unlabeled nodes using a semi-supervised classifier based on a standard GCN architecture (Kipf & Welling, 2016a). Let $\mathcal{V}_L \subset \mathcal{V}$ denote the set of labeled nodes with ground-truth labels $y$, and let $\mathcal{V}_{uL} = \mathcal{V} \setminus \mathcal{V}_L$ be the set of unlabeled nodes. For each node in $\mathcal{V}_{uL}$, we predict a pseudo-label $\tilde{y}$, while preserving the true labels in $\mathcal{V}_L$. Then, for each class $c \in C$, we construct a class-specific graph by introducing a synthetic class node connected to all nodes in $\mathcal{G}$ with missing features and predicted label $c$. Feature propagation within each class graph is then performed, with the synthetic node serving as a class-level anchor. This step promotes intra-class consistency and strengthens inter-class separation, resulting in more robust downstream representations.

To account for potential errors in pseudo-label assignments, we introduce a credibility weight based on the entropy of neighborhood label distributions. This score modulates each node's contribution to its class feature according to the consistency of local labels.

**Definition 1 (Neighborhood label information entropy).** *Let $\hat{\mathcal{N}}_i = \mathcal{N}_i \cup \{v_i\}$ denote the extended neighborhood of node $v_i$ including itself. For each class $c \in C$, with $\mathbb{1}_{(.)}$ is the indicator function,*

*we define the normalized label entropy*

$$S_i = - \left(1/\log\left(|\hat{\mathcal{N}}_i|\right)\right) \sum_{c \in C} P_i(c) \cdot \log\left(P_i(c)\right) \quad with \quad P_i(c) = \left(1/|\hat{\mathcal{N}}_i|\right) \sum_{j \in \mathcal{N}_i} \mathbb{1}_{(\tilde{y}_j = c)}. \quad (5)$$

The entropy score $S_i$ lies in the range $[0, 1]$, where lower values indicate greater label consistency in the neighborhood of node $i$, and higher values reflect uncertainty. When $S_i$ is small, the node is more representative of its assigned class and should contribute more when computing the class feature. Conversely, nodes with high entropy may not provide reliable class information and are thus down-weighted. To capture this, we assign each node a confidence weight of $1 - S_i$ and compute class-specific feature $x^\star_{(c)}$ as a weighted average of features from all nodes assigned to class $c$ by

$$x^\star_{(c)} = \left(\sum_{\tilde{y}_i = c} (1 - S_i) \cdot x_i\right) / \left(\sum_{\tilde{y}_i = c} (1 - S_i)\right), \quad (6)$$

where $x_i$ is the observed/imputed feature of node $v_i$, and $x^\star_{(c)}$ is the aggregated class feature for class $c$.

To incorporate class-specific information into node features, we introduce a virtual class node $v^{(c)}$ for each class $c$, initialized with an aggregated class feature vector $x^\star_{(c)}$. For each class, we construct a class graph $\mathcal{G}^{(c)} = (\mathcal{V}^{(c)}, \mathcal{E}^{(c)})$, where the node set $\mathcal{V}^{(c)}$ includes the class node $v^{(c)}$ and the set $\mathcal{V}^{(c)}_-$ of nodes in $\mathcal{G}$ with missing features and pseudo-label $c$. Each class graph includes self-loops for all nodes and directed edges from the class node $v^{(c)}$ to every node in $\mathcal{V}^{(c)}_-$. These edges allow class-level information to flow toward incomplete nodes, guiding feature refinement. The corresponding feature matrix $X^{(c)}$ for $\mathcal{G}^{(c)}$ is defined by $X^{(c)} = [X^{(c)}_- \; x^\star_{(c)}]^T$, wherein $X^{(c)}_- \in \mathbb{R}^{|\mathcal{V}^{(c)}_-| \times F}$ contains the features of nodes in $\mathcal{V}^{(c)}_-$ imputed during the earlier stage, and $x^\star_{(c)} \in \mathbb{R}^{1 \times F}$ is the class anchor feature assigned to the virtual node $v^{(c)}$.

In the pre-classification GCN, the final layer maps each node embedding $z_i$ to a class probability distribution $\mathbf{y}_i$. These probabilities reflect the discriminative confidence of the node across classes and can be used to guide class-level propagation. We apply a temperature-scaled softmax to control the sharpness of this distribution as $\mathbf{y}_i = \text{softmax}(z_i/T)$, where $T > 0$ is a temperature parameter. The scalar value $\mathbf{y}_i$ assigned to the predicted class serves as the self-loop weight, while $(1 - \mathbf{y}_i)$ is used for the incoming edge from the class node.

We construct a weighted adjacency matrix $\mathbf{W}^{(c)}$ to perform diffusion within each class graph $\mathcal{G}^{(c)}$. The class node is placed last in the node ordering. For each node $i$ in $\mathcal{V}^{(c)}_-$, the diagonal entry $\mathbf{W}^{(c)}_{ii}$ is set to its predicted class probability $\hat{y}_i$. The entry $\mathbf{W}^{(c)}_{i,\text{cls}}$ is set to $1 - \hat{y}_i$, allowing class-level information to flow toward uncertain nodes. The class node has a self-loop with weight 1. All other entries are zero.[1] Feature refinement is performed using a single propagation step as $\hat{X}^{(c)} = \mathbf{W}^{(c)} X^{(c)}$, where $X^{(c)}$ contains the imputed node features and the class anchor. After processing all class graphs, the refined features are mapped back to their original node indices. Observed features are then restored to produce the final output matrix $\hat{X}$, and passed to the downstream GNN for prediction.

## 3 EXPERIMENTS

### 3.1 EXPERIMENTAL SETUP

**Datasets.** We evaluate our method on five benchmark datasets: three citation networks (**Cora**, **CiteSeer**, and **PubMed**) (Sen et al., 2008), where nodes represent papers and edges indicate citation links; and Amazon co-purchase networks (**Photo** and **Computers**) (Shchur et al., 2018), where nodes are products and edges connect items frequently bought together. Additional dataset details

---

[1] For example, the matrix with two incomplete nodes reads $\mathbf{W}^{(c)} = \begin{bmatrix} \hat{y}_1 & 0 & 1 - \hat{y}_1 \\ 0 & \hat{y}_2 & 1 - \hat{y}_2 \\ 0 & 0 & 1 \end{bmatrix}$, where the class node corresponds to the last row and column.

are provided in Appendix A.3.1. To simulate missing features, we randomly remove node attributes according to a missing rate parameter $mr$, under two settings: ***Uniform Missing.*** A random $mr$ percentage of feature entries in the matrix $X$ is masked and set to zero, simulating cases where nodes have partially missing attributes; ***Structural Missing.*** A random $mr$ percentage of nodes is selected, and all features associated with those nodes are masked, modeling cases where some nodes lack features entirely.

**Baselines.** We compare against three baseline models and four state-of-the-art methods from both *deep learning-based* and *diffusion-based* approaches. **Zero (Baseline1)** sets all missing feature values to zero and applies a standard GCN (Kipf & Welling, 2016a). **PaGCN (Baseline2)** (Zhang et al., 2024) uses partial graph convolution over observed features without modeling missingness. **FP (Baseline3)** (Rossi et al., 2022) performs direct feature propagation using the normalized adjacency matrix. Among state-of-the-art methods, **GRAFENNE** (Gupta et al., 2023) constructs a three-phase message-passing framework to learn on graphs. **ITR** (Tu et al., 2022) and **ASD-VAE** (Jiang et al., 2024) utilize the distribution relationship between attributes and structures for feature completion. **PCFI** (Um et al., 2023) incorporates inter-node and inter-channel correlations through confidence-aware diffusion.

**Evaluation settings and implementation.** We evaluate FSD-CAP on semi-supervised node classification and link prediction tasks. For node classification, we follow the setup in Gasteiger et al. (2019), selecting 20 nodes per class from a random pool of 1500 nodes used for training and validation; the remaining nodes are used for testing. Classification accuracy is used to assess imputation quality. For link prediction, we adopt the edge split from Kipf & Welling (2016b), using $85\%$ of edges for training, $5\%$ for validation, and $10\%$ for testing. AUC and AP are used as evaluation metrics.

We average results over 10 random data splits and report the mean and standard deviation of accuracy, AUC, and AP. Hyperparameters are selected by grid search on the validation set. For baselines, we adopt the settings from the authors' released code or papers; when such settings are unavailable, we run a grid search over a reasonable range. Additional details are provided in Appendix A.3.5 and Appendix A.3.6.

## 3.2 SEMI-SUPERVISED NODE CLASSIFICATION

We evaluate how classification accuracy varies with the missing rate $mr$ in the semi-supervised node classification task. The missing rate is increased from 0.6 to 0.995, and all methods tested under both structural and uniform missing scenarios. Results for the Cora and CiteSeer datasets are shown in Figure 2; additional results for PubMed, Photo, and Computers are provided in Appendix A.4.1.

As $mr$ increases, the accuracy of all methods declines. Latent-space approaches (**ITR** and **ASD-VAE**) exhibit significant performance degradation at high missing rates. In contrast, **FSD-CAP** remains robust across datasets and maintains competitive accuracy even at $mr = 0.995$. Diffusion-based methods generally outperform latent-space methods, with **FSD-CAP** consistently achieving the best performance. Compared to the strongest baseline, **PCFI**, our method shows larger gains as the missing rate increases.

As shown in Table 1, we report node classification accuracy for all methods across five datasets at a fixed missing rate of $mr = 0.995$. **ITR** is designed specifically for structural missing scenarios, where

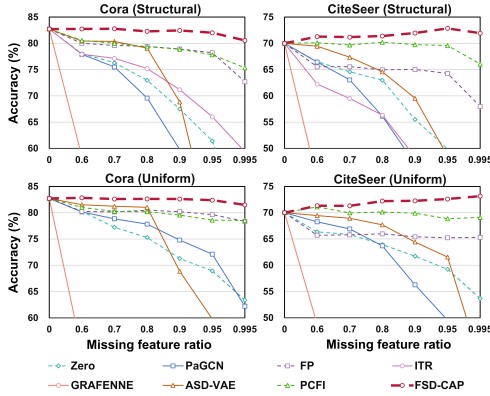

Figure 2: Node classification accuracy (%) comparison on Cora and CiteSeer datasets with $mr \in \{0.6, 0.7, 0.8, 0.9, 0.95, 0.995\}$. The top row displays results for structural missing, while the bottom row shows results for uniform missing. Methods that encounter out-of-memory errors or are not applicable to specific missing scenarios are excluded from the corresponding plots.

Table 1: Accuracy (%) comparison for node classification at $mr = 0.995$. Bold values indicate the best performance; underlined values indicate the second-best. "OOM" denotes out-of-memory errors. "-" indicates methods that are not applicable under the given missing setting.

**Structural Missing**

| Dataset | Full Features | Zero | PaGCN | FP | GRAFENNE | ITR | ASD-VAE | PCFI | FSD-CAP |
|---|---|---|---|---|---|---|---|---|---|
| Cora | 82.72 ± 1.61 | 43.50 ± 8.69 | 31.78 ± 5.68 | 72.71 ± 2.49 | 33.29 ± 5.29 | 59.37 ± 1.72 | 30.01 ± 0.55 | 75.36 ± 1.86 | **80.56 ± 1.83** |
| CiteSeer | 70.00 ± 1.35 | 31.29 ± 5.04 | 24.24 ± 1.63 | 57.98 ± 2.93 | 23.43 ± 2.37 | 33.83 ± 1.44 | 27.85 ± 3.93 | 66.06 ± 2.78 | **71.94 ± 1.32** |
| PubMed | 77.46 ± 2.09 | 46.14 ± 4.26 | 38.80 ± 5.04 | 74.18 ± 3.15 | 41.84 ± 1.71 | OOM | OOM | 74.44 ± 2.11 | **76.98 ± 1.41** |
| Photo | 91.63 ± 0.62 | 79.04 ± 2.26 | 64.31 ± 6.17 | 86.34 ± 1.09 | 50.73 ± 5.68 | 73.59 ± 3.98 | 30.85 ± 8.95 | 87.38 ± 1.04 | **89.18 ± 0.97** |
| Computers | 84.72 ± 1.25 | 71.71 ± 2.47 | 58.56 ± 2.36 | 77.19 ± 2.16 | 40.31 ± 6.26 | OOM | OOM | 78.71 ± 1.49 | **81.64 ± 1.21** |
| Average | 81.31 ± 1.38 | 54.34 ± 4.54 | 43.54 ± 4.18 | 73.68 ± 2.36 | 37.92 ± 4.26 | OOM | OOM | 76.39 ± 1.86 | **80.06 ± 1.21** |

**Uniform Missing**

| Dataset | Full Features | Zero | PaGCN | FP | GRAFENNE | ITR | ASD-VAE | PCFI | FSD-CAP |
|---|---|---|---|---|---|---|---|---|---|
| Cora | 82.72 ± 1.61 | 63.37 ± 2.02 | 62.21 ± 1.83 | 78.36 ± 1.76 | 39.86 ± 4.81 | - | 33.22 ± 5.27 | 78.55 ± 1.37 | **81.49 ± 1.95** |
| CiteSeer | 70.00 ± 1.35 | 53.66 ± 2.65 | 28.58 ± 4.31 | 65.31 ± 1.29 | 29.65 ± 2.76 | - | 41.65 ± 8.90 | 69.11 ± 1.87 | **73.15 ± 0.98** |
| PubMed | 77.46 ± 2.09 | 54.26 ± 2.68 | 41.48 ± 2.00 | 73.74 ± 2.18 | 44.43 ± 1.39 | - | OOM | 76.01 ± 1.64 | **77.46 ± 1.15** |
| Photo | 91.63 ± 0.62 | 84.96 ± 1.25 | 85.61 ± 0.69 | 88.04 ± 1.53 | 51.39 ± 5.21 | - | 32.45 ± 14.95 | 88.55 ± 1.26 | **89.40 ± 0.94** |
| Computers | 84.72 ± 1.25 | 78.99 ± 1.04 | 77.58 ± 1.96 | 80.67 ± 1.46 | 36.98 ± 2.25 | - | OOM | 81.64 ± 1.05 | **83.57 ± 0.95** |
| Average | 81.31 ± 1.38 | 67.05 ± 3.08 | 59.09 ± 2.16 | 77.22 ± 1.64 | 40.46 ± 3.28 | - | OOM | 78.77 ± 1.44 | **81.01 ± 1.19** |

nodes are either fully observed or entirely missing, and cannot be applied to uniform missing settings. We therefore report its performance only under structural missing.

Our method consistently achieves the highest accuracy across all datasets at $mr = 0.995$. In extreme missing scenarios, several deep learning-based methods such as **GRAFENNE** and **ASD-VAE** perform worse than the simple zero-filling baseline (**Baseline1**). On large-scale datasets like Photo and Computers, both **ASD-VAE** and **ITR** fail with out-of-memory errors, highlighting their limited scalability. In the structural missing setting, **FSD-CAP** outperforms the best-performing baseline (**PCFI**) with a relative improvement of 6.90% on Cora, computed as $(80.56 - 75.36)/75.36 \times 100\% = 6.90\%$. It also achieves gains of 8.90%, 3.41%, 2.06%, and 3.72% on CiteSeer, PubMed, Photo, and Computers, respectively. Under the uniform missing setting, where only 0.5% of node features are retained, **FSD-CAP** reaches average accuracy comparable to a GCN trained on fully observed features. Notably, on CiteSeer, it even surpasses the performance of the GCN with complete features.

## 3.3 LINK PREDICTION

Table 2 reports AUC and AP scores for the link prediction task on five datasets at $mr = 0.995$. Due to severe performance degradation of some methods under high missing rates, we restrict the comparison to **ITR** and the diffusion-based methods **FP** and **PCFI**. As before, **ITR** is applicable only to structural missing and fails with out-of-memory errors on large-scale datasets.

Our method consistently outperforms both **FP** and **PCFI** across all datasets and missing types, with the sole exception of the AP metric on PubMed. These results confirm that our approach remains effective under extreme feature sparsity in both classification and link prediction tasks.

Table 2: Performance comparison for link prediction at $mr = 0.995$. OOM denotes out of memory. The bold and underlined represent the best and the suboptimal performance (%), respectively.

| Dataset | Metric | Full Features | Structural Missing | | | | Uniform Missing | | |
|---|---|---|---|---|---|---|---|---|---|
| | | | ITR | FP | PCFI | FSD-CAP | FP | PCFI | FSD-CAP |
| Cora | AUC | 92.12 ± 0.71 | 82.01 ± 2.73 | 84.79 ± 1.99 | 85.94 ± 1.49 | **87.97 ± 1.24** | 87.02 ± 1.26 | 86.85 ± 1.69 | **90.01 ± 1.10** |
| | AP | 92.45 ± 0.72 | 83.76 ± 2.87 | 87.04 ± 2.46 | 87.95 ± 1.20 | **88.80 ± 1.18** | 89.17 ± 0.83 | 88.86 ± 1.15 | **90.66 ± 0.75** |
| CiteSeer | AUC | 91.02 ± 1.15 | 71.35 ± 3.25 | 81.40 ± 1.32 | 80.13 ± 1.80 | **87.48 ± 0.96** | 82.44 ± 1.50 | 82.98 ± 1.78 | **87.70 ± 1.09** |
| | AP | 91.59 ± 1.20 | 73.30 ± 2.42 | 83.62 ± 1.52 | 83.72 ± 1.58 | **87.82 ± 1.19** | 84.81 ± 0.85 | 86.11 ± 1.60 | **88.23 ± 1.16** |
| PubMed | AUC | 96.88 ± 0.20 | OOM | 86.18 ± 0.43 | 82.68 ± 0.70 | **88.39 ± 0.57** | 86.32 ± 0.21 | 84.46 ± 0.85 | **88.50 ± 0.45** |
| | AP | 97.13 ± 0.24 | OOM | 83.24 ± 0.74 | **86.03 ± 0.32** | 85.54 ± 0.72 | 83.33 ± 0.41 | **86.80 ± 0.39** | 85.60 ± 0.42 |
| Photo | AUC | 97.85 ± 0.17 | 97.11 ± 0.43 | 91.44 ± 4.58 | 96.41 ± 0.47 | **97.55 ± 0.16** | 94.97 ± 3.06 | 97.07 ± 0.19 | **98.16 ± 0.05** |
| | AP | 97.61 ± 0.22 | 96.96 ± 0.48 | 91.11 ± 4.33 | 96.02 ± 0.55 | **97.32 ± 0.25** | 94.51 ± 3.12 | 96.90 ± 0.23 | **98.08 ± 0.08** |
| Computers | AUC | 97.44 ± 0.23 | OOM | 90.01 ± 3.57 | 94.44 ± 0.34 | **96.85 ± 0.13** | 92.95 ± 3.60 | 95.59 ± 0.23 | **97.69 ± 0.07** |
| | AP | 97.35 ± 0.25 | OOM | 90.44 ± 3.10 | 94.45 ± 0.34 | **96.83 ± 0.14** | 93.08 ± 3.20 | 95.56 ± 0.29 | **97.74 ± 0.07** |
| Average | AUC | 95.06 ± 0.49 | OOM | 86.76 ± 2.38 | 87.92 ± 0.96 | **91.65 ± 0.61** | 88.74 ± 1.93 | 89.39 ± 0.95 | **92.41 ± 0.55** |
| | AP | 95.23 ± 0.53 | OOM | 87.09 ± 2.43 | 89.63 ± 0.80 | **91.26 ± 0.70** | 88.98 ± 1.68 | 90.85 ± 0.73 | **92.06 ± 0.50** |

### 3.4 ABLATION STUDY AND EVALUATION ON LARGE-SCALE AND HETEROPHILOUS DATASETS

We evaluate the contribution of each component in the framework, namely the fractional diffusion operator, progressive subgraph diffusion, and class-level feature refinement. Ablation results and discussion appear in Appendix A.2.1. We then test FSD-CAP on large-scale datasets and on datasets with heterophily. Across these settings, the method remains competitive and adapts well. Detailed results and analysis are provided in Appendix A.2.7 and Appendix A.2.8.

## 4 RELATED WORKS

### 4.1 GRAPH FEATURE IMPUTATION

Learning from incomplete data due to missing attributes or partial observations is a common challenge in real-world graph applications (Little & Rubin, 2019). Existing approaches fall into two main paradigms: *end-to-end* models, which integrate feature completion directly into the learning pipeline, and *imputation-then-training* models, which treat imputation as a separate preprocessing step (You et al., 2020; Huo et al., 2023). *End-to-end* methods aim to jointly learn node representations and impute missing features within a unified architecture. PaGCN (Zhang et al., 2024) introduces a partial graph convolution that aggregates only observed features but struggles under high missing rates. GRAFENNE (Gupta et al., 2023) constructs a three-phase message-passing framework enabling dynamic feature acquisition, though its scalability is limited. Although expressive, these methods often require significant training data and computational resources. On the other hand, *imputation-then-training* approaches estimate missing values first utilizing statistical methods (Liu et al., 2019; Batista & Monard, 2002) or learned generative models (Kingma & Welling, 2013; Yoon et al., 2018) before applying standard GNNs to the completed data. SAT (Chen et al., 2020) optimizes distributional discrepancies between graph structure and node attributes, while ITR (Tu et al., 2022) and RITR (Tu et al., 2024) refine the input by amplifying trustworthy observed features. These models benefit from modularity but remain vulnerable to bias when missing rates are high. A recent survey provides a comprehensive overview of this landscape (Xia et al., 2025).

### 4.2 GRAPH DIFFUSION

Diffusion-based graph representation learning (Gasteiger et al., 2019; Chamberlain et al., 2021) has been widely explored as a mechanism for neighborhood smoothing. Recent developments have extended classical diffusion beyond standard message-passing schemes. For example, ADC (Zhao et al., 2021) improves over GDC (Gasteiger et al., 2019) by learning adaptive neighborhood sizes from data, eliminating the need for manual tuning and enhancing both flexibility and generalization. In the context of imputation, diffusion can be viewed as a heat kernel process (Kondor & Lafferty, 2002), where observed features propagate through graph to estimate missing values. The process assumes feature homophily (i.e., neighboring nodes are likely to share similar attributes) and is often analyzed via Dirichlet energy, which decreases as smoothness increases (Zhou et al., 2021; Zhang, 2023; Zhang et al., 2023). FP (Rossi et al., 2022) formalizes this by directly minimizing Dirichlet energy, yielding a simple and effective iterative algorithm for feature diffusion under high sparsity. However, it does not incorporate feature correlations or relational context. Recent work addresses this limitation by integrating richer structural signals. PCFI (Um et al., 2023) guides diffusion using feature-level confidence across nodes and channels, while SGHFP (Lei et al., 2023) models higher-order relationships using a hypergraph structure (Dong et al., 2025a). Nonetheless, most existing methods apply uniform diffusion across nodes, ignoring variations in completion difficulty. As a result, poorly estimated nodes may contaminate their neighbors.

## 5 CONCLUSION

In this paper, we propose a novel diffusion-based feature imputation approach for incomplete graph learning. By introducing a fractional diffusion operator and distance-guided progressive subgraph diffusion, we are able to adjust the propagation sharpness according to local graph structure and propagate information from observed to missing nodes hierarchically, thereby improving the stability of the diffusion process and the reliability of the imputed features. We further design a class-

level refinement mechanism to enhance feature quality under severe feature sparsity. Extensive experiments conducted on a variety of benchmark datasets demonstrate that our method not only consistently outperforms state-of-the-art approaches in semi-supervised node classification and link prediction tasks, with particularly notable improvements under extreme feature missingness, but also exhibits remarkable adaptability to diverse graph structures.

## ETHICS STATEMENT

FSD-CAP improves the robustness of GNNs under high feature-missing rates and may benefit applications in domains such as social networks and recommender systems. However, as with any imputation technique, there is a risk of misuse, particularly in inferring sensitive or private attributes from partial data. We encourage responsible use, including proper access controls and ethical oversight, especially when applying the model to contexts involving personal or sensitive information.

## REPRODUCIBILITY STATEMENT

The proposed imputation framework is introduced in detail in Section 2, with a summary of the experimental setup provided in Section 3. Further details, including data splitting strategies and hyperparameter settings, are given in the appendix. For reproducibility, we make the code and corresponding random seeds publicly available.

## ACKNOWLEDGMENTS

This work was supported by grants from the Natural Science Foundation of Shanxi Province (2024JCJCQN-66), Science and Technology Commission of Shanghai Municipality (NO.24511106900), Key R&D Program of Zhejiang (2024SSYS0091).

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

# A   APPENDIX

OUTLINE OF THE SUPPLEMENTARY MATERIAL

**Section 1 – Proofs of main results:** Formal proofs for the theoretical results stated in the main paper.

**Section 2 – Experimental analyses:** We present additional evaluations to support the effectiveness and robustness of FSD-CAP. First, ablation studies show that removing any single component leads to a consistent drop in performance across all datasets. Second, sensitivity analyses demonstrate the stability of the framework with respect to key parameters. A comparison with FP, a global diffusion baseline, highlights FSD-CAP's ability to maintain accuracy at greater distances from observed nodes, validating the progressive diffusion mechanism.

We also analyze class-level characteristics of imputed features and visualize the outputs of FP and PCFI to show that FSD-CAP better preserves inter-class separability. We further validate FSD-CAP's strong practical scope and generality through comparative experiments on large-scale and heterophily datasets. Finally, we report classification accuracy under missing rates from $60\%$ to $99.5\%$, showing that FSD-CAP is highly robust to both the rate and structure of missing data. Under uniform missing with $mr = 0.995$, the average performance drop is just $0.3\%$ relative to the fully observed case; in some cases, performance even improves. FSD-CAP consistently achieves higher node classification accuracy than the state-of-the-art baseline PCFI under varying levels of feature missingness.

**Section 3 – Implementation and hyperparameters:** Experimental setup for both node classification and link prediction tasks, including dataset statistics, data splits, evaluation metrics, model architectures, and hyperparameter configurations for all baselines and FSD-CAP.

**Section 4 – Supplementary figures and tables:** Additional results comparing node classification accuracy on PubMed, Photo, and Computers across multiple missing rates ($mr \in 0.6, 0.7, 0.8, 0.9, 0.95, 0.995$). These results extend Section 3.2 of the main paper and further validate the robustness of FSD-CAP.

**Section 5 – Declaration of LLM usage:** We make a full disclosure regarding the utilization of the large language model (LLM) throughout the process of completing this thesis.

**Section 6 – Pseudocode of FSD-CAP:** We provide the pseudocode for Fractional Subgraph Diffusion (Algorithm 1) and Class-Aware Propagation (Algorithm 2).

## A.1   PROOFS OF MAIN RESULTS

**Proposition A.1** (**Limiting behavior of $\mathbf{A}^\gamma$**)**.** *Let $\mathbf{A}$ be the symmetric normalized adjacency matrix of a connected graph, and $\mathbf{A}^\gamma \in \mathbb{R}^{N \times N}$ is defined as*

$$\mathbf{A}^\gamma_{ij} := (\mathbf{A}_{ij})^\gamma / \left( \sum_{k=1}^{N} (\mathbf{A}_{ik})^\gamma \right). \tag{7}$$

*We have*

$$\lim_{\gamma \to 0^+} \mathbf{A}^\gamma_{ij} = \begin{cases} \dfrac{1}{|\mathcal{N}(i)|} & \text{if } \mathbf{A}_{ij} > 0 \\ 0 & \text{otherwise} \end{cases}, \qquad \lim_{\gamma \to \infty} \mathbf{A}^\gamma_{ij} = \begin{cases} 1 & \text{if } j \in \arg\max_k \mathbf{A}_{ik} \\ 0 & \text{otherwise} \end{cases}. \tag{8}$$

*where $\mathcal{N}(i) = \{j \mid \mathbf{A}_{ij} > 0\}$ denotes the set of neighbors of node $i$.*

*Proof.* Let $i$ be a fixed row index. By definition, the entries of $\mathbf{A}^\gamma$ are given by

$$\mathbf{A}^\gamma_{ij} = \frac{\mathbf{A}^\gamma_{ij}}{\sum_{k \in \mathcal{N}(i)} \mathbf{A}^\gamma_{ik}}, \tag{9}$$

where the sum is taken over the neighborhood $\mathcal{N}(i) = k \mid \mathbf{A}_{ik} > 0$. Since $\mathbf{A}$ is a symmetric normalized adjacency matrix, all its entries are nonnegative and satisfy $\mathbf{A}_{ij} \in [0, 1]$.

We consider two limiting cases:

**Case 1: $\gamma \to 0^+$.** For all $j \in \mathcal{N}(i)$, we have $\mathbf{A}_{ij}^{\gamma} \to 1$ as $\gamma \to 0^+$. Therefore, the numerator tends to 1 for each neighbor, and the denominator tends to $|\mathcal{N}(i)|$. As a result,

$$\lim_{\gamma \to 0^+} \mathbf{A}_{ij}^{\gamma} = \frac{1}{|\mathcal{N}(i)|} \quad \text{if } j \in \mathcal{N}(i), \tag{10}$$

and zero otherwise. Thus, the row converges to a uniform distribution over the neighbors of node $i$.

**Case 2: $\gamma \to \infty$.** In this case, the exponentiation amplifies differences between entries. Specifically, the largest value $\mathbf{A}_{ik}$ dominates the sum in the denominator, and the softmax approaches a one-hot distribution centered at the maximum. Let $j^{\star} \in \arg\max_k \mathbf{A}_{ik}$. Then,

$$\lim_{\gamma \to \infty} \mathbf{A}_{ij}^{\gamma} = \begin{cases} 1 & \text{if } j = j^{\star} \\ 0 & \text{otherwise} \end{cases}, \tag{11}$$

which completes the proof. $\qquad\square$

**Theorem A.1 (Fractional diffusion on feature propagation).** *Let $\mathbf{A}$ and $\mathbf{A}^{\gamma}$ denote the lazy transition and fractional diffusion operator as defined in equation 7, respectively. For any feature matrix $X \in \mathbb{R}^{N \times F}$, define the propagated features by $X^{\gamma} := \mathbf{A}^{\gamma} X$. Then for each node $i$, we have*

$$\lim_{\gamma \to 0^+} X_{[i,:]}^{\gamma} = \frac{1}{|\mathcal{N}(i)|} \sum_{j \in \mathcal{N}(i)} X_{[j,:]} \quad \text{and} \quad \lim_{\gamma \to \infty} X_{[i,:]}^{\gamma} = X_{[j^{\star},:]},$$

*where $\mathcal{N}(i) = \{j \mid \mathbf{A}_{ij} > 0\}$ the neighborhood of $v_i$, and $j^{\star} \in \arg\max_k \mathbf{A}_{ik}$ is the index of its strongest neighbor.*

*Proof.* The propagated feature vector at node $i$ is defined by

$$X_{[i,:]}^{\gamma} = \sum_{j=1}^{N} \mathbf{A}_{ij}^{\gamma} X_{[j,:]},$$

the limiting regimes of $\gamma$ thus read

**Case 1: $\gamma \to 0^+$.** As $\gamma$ approaches zero, we have $\mathbf{A}_{ij}^{\gamma} \to 1/|\mathcal{N}(i)|$ for all $j \in \mathcal{N}(i)$, and zero otherwise. Therefore,

$$X_{[i,:]}^{\gamma} \to \frac{1}{|\mathcal{N}(i)|} \sum_{j \in \mathcal{N}(i)} X_{[j,:]},$$

which corresponds to uniform averaging over the neighborhood of node $i$.

**Case 2: $\gamma \to \infty$.** The largest entry in row $i$ of $\mathbf{A}$ dominates the normalization. Specifically, if $j^{\star} \in \arg\max_k \mathbf{A}_{ik}$, then $\mathbf{A}_{ij}^{\gamma} \to 1$ if $j = j^{\star}$, and zero otherwise. Hence,

$$X_{[i,:]}^{(\gamma)} \to X_{[j^{\star},:]},$$

which corresponds to selecting the feature vector of the strongest neighbor. $\qquad\square$

**Theorem A.2 (Convergence of subgraph diffusion).** *Let $A^{(m)}$ be the adjacency matrix of the $m$-th layer subgraph $\mathcal{G}^{(m)}$, and let $\mathbf{A}^{\gamma,m}$ denote its fractional diffusion matrix as defined in equation 7. Let $x^{(m)} \in \mathbb{R}^{|\mathcal{V}^{(m)}|}$ be the feature vector in channel $\ell$, where missing entries are initialized to zero. Let $\lambda > 0$ denote the retention coefficient, and $M$ be the binary mask vector for observed entries in channel $\ell$. Define the update sequence by*

$$x^{(m)}(t) = x^{(m)}(0) \odot M + \left(\mathbf{A}^{\gamma,m} x^{(m)}(t-1) + \lambda x^{(m-1)}(K)\right) \odot (1 - M), \quad t = 1, \dots, K$$

*Then, for sufficiently large $K$, the sequence $x^{(m)}(t)$ converges to a fixed (unique) state.*

*Proof.* Fix a feature channel $\ell$, and consider the $m$-th subgraph layer $\mathcal{G}^{(m)}$ with $n = |\mathcal{V}^{(m)}|$ nodes. Let $x^{(m)}$ denotes the vector of $\ell$-th feature values across all nodes in this subgraph

$$x^{(m)} = \begin{bmatrix} x_1^{(m)}[\ell] \\ x_2^{(m)}[\ell] \\ \vdots \\ x_n^{(m)}[\ell] \end{bmatrix}.$$

Recall $M \in \{0,1\}^{N \times F}$ be the full binary feature mask. Define the channel-wise mask vector $M^{(\ell)} \in \{0,1\}^n$ by $M_i^{(\ell)} := M[i, \ell]$, which indicates whether node $v_i$'s feature in channel $\ell$ is observed.

Let $\mathbf{A}^{(m)}$ be the symmetric normalized adjacency matrix of the subgraph $\mathcal{G}^{(m)}$, and define the fractional diffusion operator $\mathbf{A}^{\gamma, m} \in \mathbb{R}^{n \times n}$ as

$$\mathbf{A}_{ij}^{\gamma, m} = \frac{(\mathbf{A}_{ij}^{(m)})^\gamma}{\sum_{k=1}^n (\mathbf{A}_{ik}^{(m)})^\gamma}.$$

Since $\gamma > 0$, this defines a row-stochastic matrix with nonnegative entries. Its support coincides with that of $\mathbf{A}^{(m)}$, and thus matches the topology of the subgraph. If $\mathcal{G}^{(m)}$ is connected, then $\mathbf{A}^{\gamma, m}$ is the transition matrix of an irreducible and aperiodic Markov chain.

From standard results in Markov chain theory, for any initial vector $v(0)$, the iterates

$$v(t) = (\mathbf{A}^{\gamma, m})^t v(0)$$

converge to the unique stationary distribution.

Now consider the masked and retained update. Define the diagonal projection matrices

$$P := \text{diag}(M^{(\ell)}), \quad Q := I - P = \text{diag}(1 - M^{(\ell)}).$$

Then the update rule can be written as

$$x^{(m)}(t) = Px^{(m)}(0) + Q\left(\mathbf{A}^{\gamma, m} x^{(m)}(t-1) + \lambda x^{(m-1)}(K)\right),$$

where $\lambda > 0$ is the retention factor and $x^{(m-1)}(K)$ is the converged estimate from the previous layer. This is a linear, inhomogeneous recurrence of the form

$$x^{(m)}(t) = Gx^{(m)}(t-1) + b,$$

with

$$G := Q\mathbf{A}^{\gamma, m}, \quad b := Px^{(m)}(0) + \lambda Q x^{(m-1)}(K).$$

Since $\mathbf{A}^{\gamma, m}$ is row-stochastic and $Q$ is diagonal with entries in $[0, 1]$, the matrix $G$ is sub-stochastic – its rows have nonnegative entries and sum to at most 1. Furthermore, any row corresponding to an observed feature (i.e., $M_i^{(\ell)} = 1$) satisfies $Q_{ii} = 0$, so the corresponding row of $G$ is all zeros. If at least one feature is observed (i.e., $M^{(\ell)} \neq 0$), then $G$ has at least one row with sum strictly less than 1, and thus its spectral radius satisfies $\rho(G) < 1$.

Since $\rho(G) < 1$, the recurrence converges to a unique fixed point, i.e.,

$$x^{(m)}(\infty) = (I - G)^{-1} b.$$

This proves that the update converges for each feature channel $\ell$, in every subgraph layer $m$, as long as the underlying subgraph is connected and $\gamma < \infty$. $\qquad \square$

**Theorem A.3** (**Global convergence via progressive subgraph expansion**). *Let $\mathcal{G}^{(m)}$ be the $m$-hop expansion of the observed node set $\mathcal{V}_+^\ell$ in channel $\ell$, and let $x^{(m)}$ be the corresponding feature estimate after applying the masked fractional diffusion update defined in Theorem A.2. Assume the graph $\mathcal{G}$ is connected and the diffusion sharpness parameter $\gamma$ is finite. Then as $m \to M_{\max}$, where $\mathcal{G}^{(m)} \to \mathcal{G}$, the final estimate $x^{(m)}(\infty)$ converges to the steady-state solution of the full-graph diffusion update.*

*Proof.* Fix a feature channel $\ell$, and consider sequence of subgraphs $\{\mathcal{G}^{(m)}\}$ defined by expanding the observed node set $\mathcal{V}_+^\ell$ in increasing $m$-hop neighborhoods. Each subgraph $\mathcal{G}^{(m)} = (\mathcal{V}^{(m)}, \mathcal{E}^{(m)})$ satisfies

$$\mathcal{V}^{(m)} \subseteq \mathcal{V}^{(m+1)}, \quad \mathcal{E}^{(m)} \subseteq \mathcal{E}^{(m+1)}, \quad \text{and} \quad \bigcup_m \mathcal{G}^{(m)} = \mathcal{G}.$$

Let $x^{(m)}(\infty)$ denote the steady-state solution obtained by applying the masked and retained update on $\mathcal{G}^{(m)}$, which exists by Theorem A.2. Since each update only modifies nodes in $\mathcal{V}^{(m)}$, and each $\mathcal{V}^{(m)}$ is strictly contained in the next, the sequence $\{x^{(m)}(\infty)\}_m$ defines an expanding approximation of the solution on the full graph.

We now define the full-graph update. Let $\mathcal{G} = (\mathcal{V}, \mathcal{E})$ be the complete graph, and let $\mathbf{A}^\gamma \in \mathbb{R}^{N \times N}$ be the full fractional diffusion matrix defined over all $N$ nodes

$$\mathbf{A}_{ij}^\gamma = \frac{(\mathbf{A}_{ij})^\gamma}{\sum_{k=1}^N (\mathbf{A}_{ik})^\gamma},$$

where $\mathbf{A}$ is the symmetric normalized adjacency matrix of $\mathcal{G}$. Let $x^\star$ be the solution to the global masked and retained update

$$x^\star = Px(0) + Q\left(\mathbf{A}^\gamma x^\star + \lambda x^{\text{prev}}\right),$$

where $P = \text{diag}(M^{(\ell)})$, $Q = I - P$, and $x(0)$ is the initial feature vector for this channel.

Now, observe that for any fixed node $i$, there exists a minimal radius $m_i$ such that $i \in \mathcal{V}^{(m)}$ for all $m \geq m_i$. Beyond this point, the update for $x_i^{(m)}(\infty)$ is computed using the same local topology, diffusion weights, and mask structure as in the full graph. Hence, for each node $i$, the local solution on $\mathcal{G}^{(m)}$ matches the restriction of the global solution, up to boundary conditions that vanish as the subgraph expands.

Furthermore, the retention mechanism ensures that values propagated from earlier layers remain stable and reinforce prior estimates. Because the number of nodes is finite and each node is eventually included in some $\mathcal{G}^{(m)}$, we conclude that

$$\lim_{m \to M_{\max}} x^{(m)}(\infty) = x^\star,$$

where $x^\star$ is the fixed point of the global masked diffusion update. $\square$

## A.2 EXPERIMENTAL ANALYSES:

### A.2.1 ABLATION STUDY

To evaluate the contribution of each component in FSD-CAP, we perform ablation studies under the structural missing setting with a 99.5% missing rate. Experiments are conducted on both semi-supervised node classification and link prediction tasks across all datasets. We compare the full model to the following variants, each with one component removed

1. **w/o_Frac_oper:** Removes the fractional diffusion operator.
2. **w/o_Prog_subg:** Removes the progressive subgraph diffusion.
3. **w/o_Class_refine:** Removes the class-level feature refinement.

The results of the ablation studies are reported in Table 3 for node classification and Table 4 for link prediction. In both tasks, removing any single component from FSD-CAP results in a consistent drop in performance across all datasets, confirming the contribution of each module. The performance degradation is especially pronounced in the link prediction setting.

**w/o_Frac_oper.** Removing the fractional diffusion operator results in an average accuracy drop of 2.02% in node classification, with a larger effect observed on denser co-purchase datasets such as Photo and Computers (Table 3). Its impact is even more pronounced in link prediction, where the average AUC decreases by 5.60% across five datasets (Table 4). These results highlight the

role of the fractional operator in adapting diffusion sharpness to local structure and demonstrate its importance in both tasks.

**w/o_Prog_subg.** Eliminating the progressive subgraph diffusion mechanism results in a smaller average performance loss compared to the other two components. However, as shown in Table 4, it leads to noticeable degradation on link prediction for datasets such as Cora and CiteSeer. The progressive expansion strategy helps reduce error accumulation during diffusion by gradually incorporating nodes, improving stability and overall performance.

**w/o_Class_refine.** The class-level refinement has the most significant impact on node classification performance (Table 3). Its removal particularly affects sparse graphs with multiple classes, such as Cora (7 classes) and CiteSeer (6 classes). By constructing class-specific anchor nodes and propagating within class-specific graphs, this component enhances inter-class separation and strengthens the discriminative power of imputed features.

Together, these results demonstrate that all three components play complementary roles. Their combination is essential for achieving robust and accurate imputation under extreme sparsity.

Table 3: Ablation study on semi-supervised node classification. Best values are in bold.

|  | Cora | CiteSeer | PubMed | Photo | Computers | Average |
|---|---|---|---|---|---|---|
| FSD-CAP | **80.56%** | **71.94%** | **76.98%** | **89.18%** | **81.64%** | **80.06%** |
| w/o_Frac_oper | 79.95% (-0.61%) | 71.82% (-0.12%) | 75.54% (-1.44%) | 86.27% (-2.91%) | 76.60% (-5.04%) | 78.04% (-2.02%) |
| w/o_Prog_subg | 79.34% (-1.22%) | 70.84% (-1.10%) | 75.97% (-1.01%) | 88.07% (-1.11%) | 79.14% (-2.50%) | 78.67% (-1.39%) |
| w/o_Class_refine | 76.43% (-4.13%) | 68.65% (-3.29%) | 75.52% (-1.46%) | 88.10% (-1.08%) | 78.06% (-3.58%) | 77.35% (-2.71%) |

Table 4: Ablation study on link prediction, with best performance highlighted in bold.

|  | Metric | Cora | CiteSeer | PubMed | Photo | Computers | Average |
|---|---|---|---|---|---|---|---|
| FSD-CAP | AUC | **87.97%** | **87.48%** | **88.39%** | **97.55%** | **96.85%** | **91.65%** |
|  | AP | **88.80%** | **87.82%** | **85.54%** | **97.32%** | **96.83%** | **91.26%** |
| w/o_Frac_oper | AUC | 80.54% (-7.43%) | 79.07% (-8.41%) | 85.32% (-3.07%) | 95.94% (-1.61%) | 89.39% (-7.46%) | 86.05% (-5.60%) |
|  | AP | 82.65% (-6.15%) | 81.74% (-6.08%) | 84.81% (-0.73%) | 95.46% (-1.86%) | 89.99% (-6.84%) | 86.93% (-4.33%) |
| w/o_Prog_subg | AUC | 82.67% (-5.30%) | 79.71% (-7.77%) | 86.10% (-2.29%) | 96.47% (-1.08%) | 95.75% (-1.10%) | 88.14% (-3.51%) |
|  | AP | 84.48% (-4.32%) | 80.67% (-7.15%) | 84.71% (-0.83%) | 96.16% (-1.16%) | 95.68% (-1.15%) | 88.34% (-2.92%) |
| w/o_Class_refine | AUC | 83.00% (-4.97%) | 79.58% (-7.90%) | 84.90% (-3.49%) | 96.49% (-1.06%) | 94.71% (-2.14%) | 87.74% (-3.91%) |
|  | AP | 84.69% (-4.11%) | 80.76% (-7.06%) | 83.00% (-2.54%) | 96.26% (-1.06%) | 94.63% (-2.20%) | 87.87% (-3.39%) |

### A.2.2 PARAMETER ANALYSIS

To assess the sensitivity of FSD-CAP to its key parameters ($\gamma$, $\lambda$, and $T$), we conduct controlled experiments on both semi-supervised node classification and link prediction. We use the structural missing setting with a $99.5\%$ feature missing rate, evaluating on Cora and the denser Photo dataset.

In the fractional subgraph diffusion (FSD) stage, information is propagated progressively from observed to unobserved nodes by expanding the subgraph. The fractional exponent $\gamma$ controls the sharpness of diffusion: higher values lead to more localized propagation. The retention coefficient $\lambda$ determines the weight of previous estimates in the update. The temperature $T$ is used in the class-aware refinement stage to modulate the influence of class-level anchors.

We examine the joint effect of the fractional diffusion parameter $\gamma$ and the retention coefficient $\lambda$ on model performance, with temperature $T$ fixed. Figure 3 shows classification accuracy on Cora and Photo as these parameters vary. On Cora, $\gamma$ is swept from $0.6$ to $1.6$ (step size $0.2$), while on Photo it ranges from $2.0$ to $4.0$ with the same step. In both cases, $\lambda$ is varied from $0$ to $1$ in increments of $0.1$. Results indicate that Cora achieves higher accuracy when $\lambda \in [0.2, 0.4]$ and $\gamma \in [1.2, 1.4]$, suggesting that incorporating estimates from previous diffusion layers improves imputation quality. On the denser Photo dataset, accuracy improves when $\gamma$ is set around $2.6$ to $3.0$, reflecting the benefit of sharper, edge-concentrated diffusion in dense graphs.

Figures 4 and 5 present AUC and AP scores for link prediction on Cora and Photo, respectively. The trends are consistent with those observed in node classification, confirming that performance

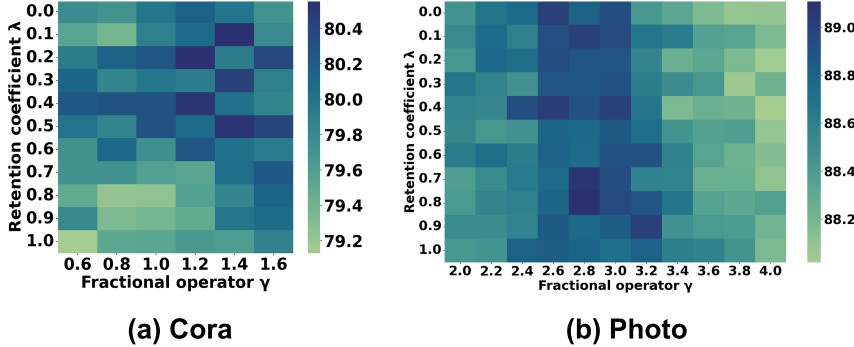

Figure 3: Sensitivity of node classification accuracy to the retention coefficient $\lambda$ and fractional exponent $\gamma$ on Cora and Photo datasets.

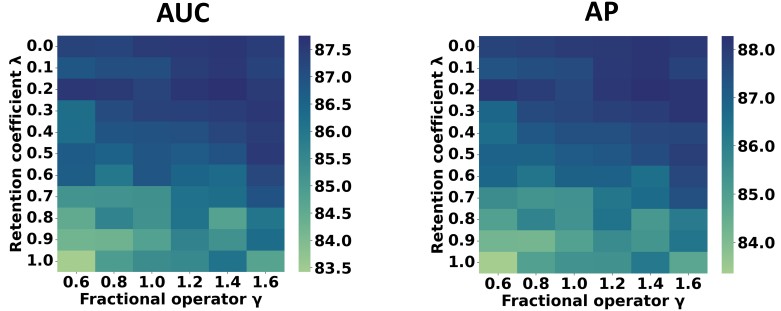

Figure 4: Sensitivity of link prediction performance to the retention coefficient $\lambda$ and fractional exponent $\gamma$ on the Cora dataset.

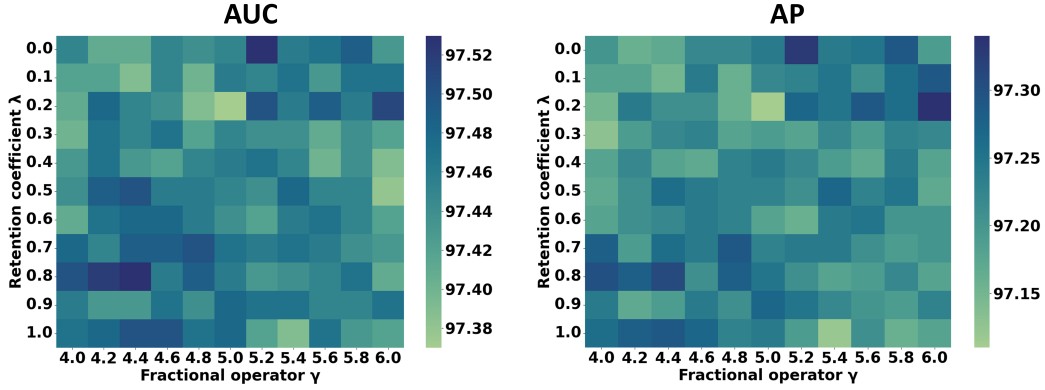

Figure 5: Sensitivity of link prediction performance to the retention coefficient $\lambda$ and fractional exponent $\gamma$ on the Photo dataset.

is sensitive to the choice of $\lambda$ and $\gamma$. On Cora, smaller values of $\lambda$ yield better results. On Photo, higher values of $\gamma$ improve performance more noticeably than in the classification setting, suggesting that localized diffusion captures structure relevant to link formation. Modulating diffusion strength across local neighborhoods allows FSD-CAP to better exploit available information. These sensitivity results further validate the role of each component in FSD-CAP. The retention mechanism stabilizes the diffusion process by preserving accurate estimates from earlier layers, reducing error propagation. The fractional operator adjusts the relative influence of neighboring nodes based on local graph structure, helping to mitigate over-smoothing and improve overall imputation quality.

In the class-aware propagation (CAP) step, a GCN-based classifier assigns pseudo-labels to unlabeled nodes. These labels are used to construct class-specific features and class-wise graphs for targeted feature refinement. The temperature parameter $T$ is applied to the softmax function to control the sharpness of the output distribution. Lower values of $T$ produce sharper, more confident predictions; higher values yield smoother distributions. When $T = 1$, the standard softmax is recovered.

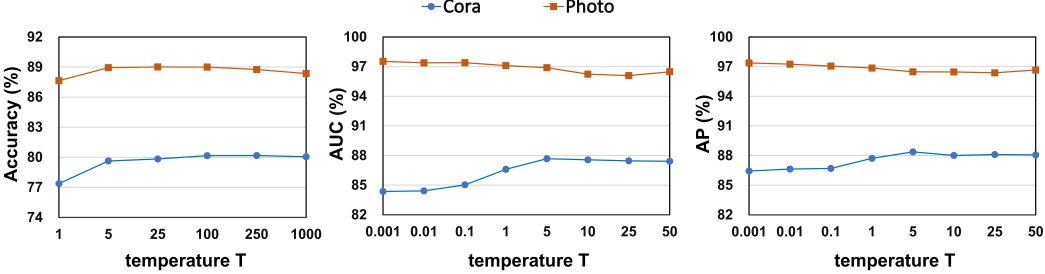

Figure 6: Sensitivity of node classification and link prediction performance to temperature parameter $T$ on the Cora and Photo datasets.

Figure 6 shows the effect of the temperature parameter $T$ on model performance across accuracy, AUC, and AP metrics for the Cora and Photo datasets. On Cora, both node classification and link prediction achieve optimal performance near $T = 5$. On the denser Photo dataset, the optimal value of $T$ differs by task: $T \approx 25$ yields the best node classification accuracy, while $T = 0.001$ gives the highest link prediction scores. At low $T$, the pseudo-label distribution becomes sharply peaked, resulting in high-confidence weights during class-aware propagation. This reduces the influence of class-level features and increases the emphasis on individual node characteristics. For link prediction on dense graphs, this behavior is beneficial, as rich local structure can guide accurate edge inference. In contrast, node classification benefits from smoother pseudo-labels (higher $T$), which reduce intra-class noise and improve class-level separation. These results highlight the importance of tuning $T$: relative to the default $T = 1$, appropriate values can significantly improve performance on both tasks.

### A.2.3 ACCURACY COMPARISON ON DIFFERENT DISTANCE

To assess how distance from observed features impacts classification accuracy, we evaluate performance under the structural missing setting with a $99.5\%$ missing rate. Nodes are grouped by their shortest-path distance to the nearest node with observed features, and average accuracy is computed for each group. We compare FSD-CAP to the global diffusion baseline FP across five datasets. Figures 7 and 8 present accuracy as a function of distance for sparse and dense graphs, respectively.

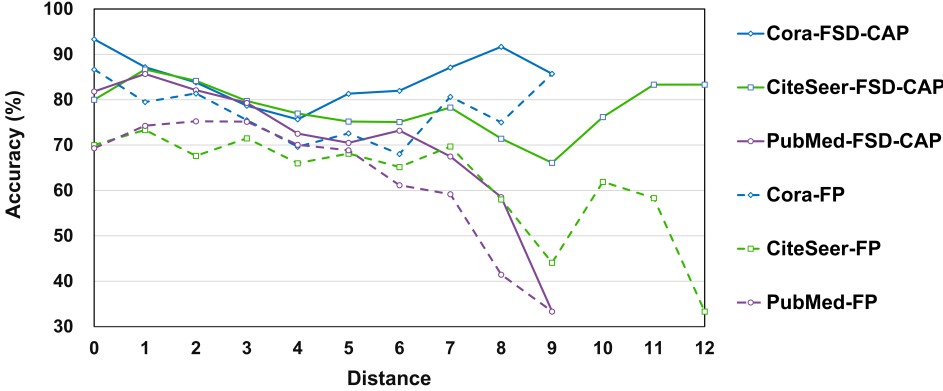

Figure 7: Node classification accuracy (%) of FP and FSD-CAP grouped by shortest-path distance to the nearest node with observed features, evaluated on Cora, CiteSeer, and PubMed.

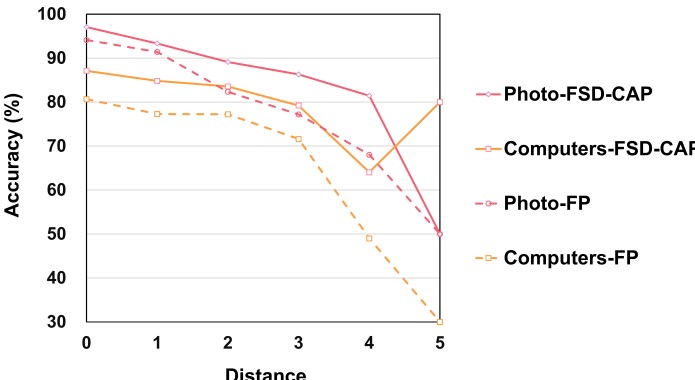

Figure 8: Node classification accuracy (%) of FP and FSD-CAP as a function of shortest-path distance to the nearest observed node, evaluated on the Photo and Computers datasets.

As shown in Figure 8, the Photo and Computers datasets are relatively dense, with most missing-feature nodes located within five hops of observed nodes. In contrast, the sparser datasets—Cora, CiteSeer, and PubMed (Figure 7)—exhibit much larger maximum distances between missing and observed nodes. Across all datasets, both FP and FSD-CAP show a general decline in accuracy as distance increases. In some cases, accuracy temporarily rises at longer distances. This is explained by the node distribution: most nodes lie within a few hops of observed regions, while distant nodes are few (Figure 9), which inflates the average accuracy of these small groups.

This trend confirms that diffusion-based methods perform better for nodes near observed features. Compared to FP, which applies global propagation, FSD-CAP reduces accuracy degradation at greater distances and improves performance for nearby nodes. These results support the effectiveness of progressive subgraph diffusion, which focuses propagation on reliable local neighborhoods and limits error accumulation, leading to more stable and accurate imputation.

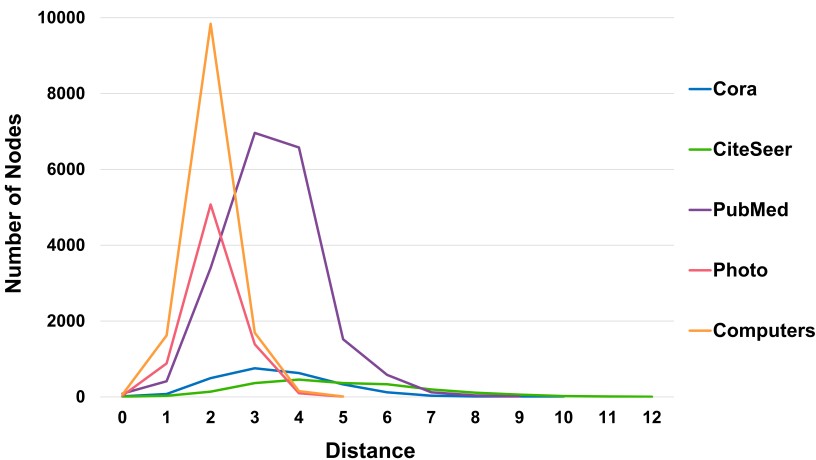

Figure 9: Number of nodes at each shortest-path distance from the nearest node with observed features across the five datasets.

### A.2.4 CLASS-LEVEL FEATURE SIMILARITY ANALYSIS

To analyze the class-level structure of features imputed by FSD-CAP, we perform experiments under the structural missing setting with a 99.5% missing rate. Cosine similarity is used to measure both intra-class and inter-class feature similarity. We compute these metrics on the imputed features and compare them to those obtained from the original, fully observed features.

Tables 5 and 6 report intra-class and inter-class cosine similarity for the original and imputed features, respectively. "Average" denotes the mean intra-class similarity across all classes. The "Ratio"

Table 5: Inter-class and intra-class cosine similarities of original features.

| Dataset | Inter-class | Intra-class | | | | | | | | | | Average | Ratio |
|---|---|---|---|---|---|---|---|---|---|---|---|---|---|
| | | class1 | class2 | class3 | class4 | class5 | class6 | class7 | class8 | class9 | class10 | | |
| Cora | 0.054 | 0.087 | 0.117 | 0.092 | 0.070 | 0.071 | 0.089 | 0.116 | - | - | - | 0.091 | 1.70 |
| CiteSeer | 0.042 | 0.053 | 0.063 | 0.063 | 0.067 | 0.078 | 0.061 | - | - | - | - | 0.064 | 1.54 |
| PubMed | 0.063 | 0.112 | 0.094 | 0.078 | - | - | - | - | - | - | - | 0.094 | 1.51 |
| Photo | 0.337 | 0.282 | 0.495 | 0.331 | 0.435 | 0.354 | 0.381 | 0.289 | 0.354 | - | - | 0.365 | 1.08 |
| Computers | 0.348 | 0.371 | 0.347 | 0.442 | 0.390 | 0.300 | 0.449 | 0.534 | 0.455 | 0.393 | 0.414 | 0.409 | 1.17 |

Table 6: Inter-class and intra-class cosine similarities of features imputed by FSD-CAP under the structural missing setting with a missing rate of $99.5\%$.

| Dataset | Inter-class | Intra-class | | | | | | | | | | Average | Ratio |
|---|---|---|---|---|---|---|---|---|---|---|---|---|---|
| | | class1 | class2 | class3 | class4 | class5 | class6 | class7 | class8 | class9 | class10 | | |
| Cora | 0.827 | 0.937 | 0.986 | 0.992 | 0.907 | 0.828 | 0.887 | 0.950 | - | - | - | 0.926 | 1.12 |
| CiteSeer | 0.671 | 0.854 | 0.817 | 0.947 | 0.868 | 0.962 | 0.958 | - | - | - | - | 0.901 | 1.34 |
| PubMed | 0.905 | 0.930 | 0.961 | 0.938 | - | - | - | - | - | - | - | 0.943 | 1.04 |
| Photo | 0.884 | 0.983 | 0.984 | 0.987 | 0.992 | 0.995 | 0.978 | 0.991 | 0.992 | - | - | 0.988 | 1.12 |
| Computers | 0.868 | 0.994 | 0.867 | 0.982 | 0.923 | 0.928 | 0.979 | 0.973 | 0.966 | 0.920 | 0.975 | 0.946 | 1.09 |

is defined as the average intra-class similarity divided by the inter-class similarity. A ratio greater than 1 indicates better class separability.

As shown in Table 5, the original features consistently yield higher intra-class similarity than inter-class similarity across all datasets, confirming their strong class-discriminative structure. Table 6 shows that FSD-CAP preserves this structure even under $99.5\%$ feature missing. In all cases, the ratio remains above 1, indicating that the imputed features retain meaningful class-level separation.

These results confirm that FSD-CAP maintains discriminative feature geometry under extreme sparsity. As shown in Table 11 (Appendix A.2.9), the resulting classification performance is comparable to, and in some cases exceeds, that of the fully observed setting.

### A.2.5 T-SNE VISUALIZATION

Under the structural missing setting with a $99.5\%$ feature missing rate, we reconstruct node features using two diffusion-based baselines (FP and PCFI) and our proposed FSD-CAP. We then apply t-SNE to project the imputed features into two dimensions for qualitative comparison.

Figures 10 and 11 show the t-SNE visualizations across five datasets, with nodes colored by ground-truth class labels. FSD-CAP produces more compact and well-separated clusters than FP and PCFI. Nodes from the same class form cohesive groups with clearer boundaries between classes. These visual patterns align with the similarity analysis in Appendix A.2.4, further demonstrating FSD-CAP's ability to preserve class structure and support downstream tasks such as node classification.

### A.2.6 COMPLEXITY ANALYSIS

**Computational complexity.** FSD-CAP consists of two stages: fractional subgraph diffusion (FSD) and class-aware propagation (CAP).

To make the cost concrete, we report wall-clock time (in seconds), including both the time for feature imputation and training of downstream GCN for FSD-CAP and baseline models in the Table 7. We measure a single split per dataset under a structural missing setting with a 99.5% missing rate. All experiments are run on a 40-core Intel Xeon Silver 4210R CPU with one NVIDIA RTX 4090 GPU.

As shown in the table, deep learning-based methods incur significantly higher computational costs compared to diffusion-based imputation approaches. For FSD-CAP, the vast majority of the time is

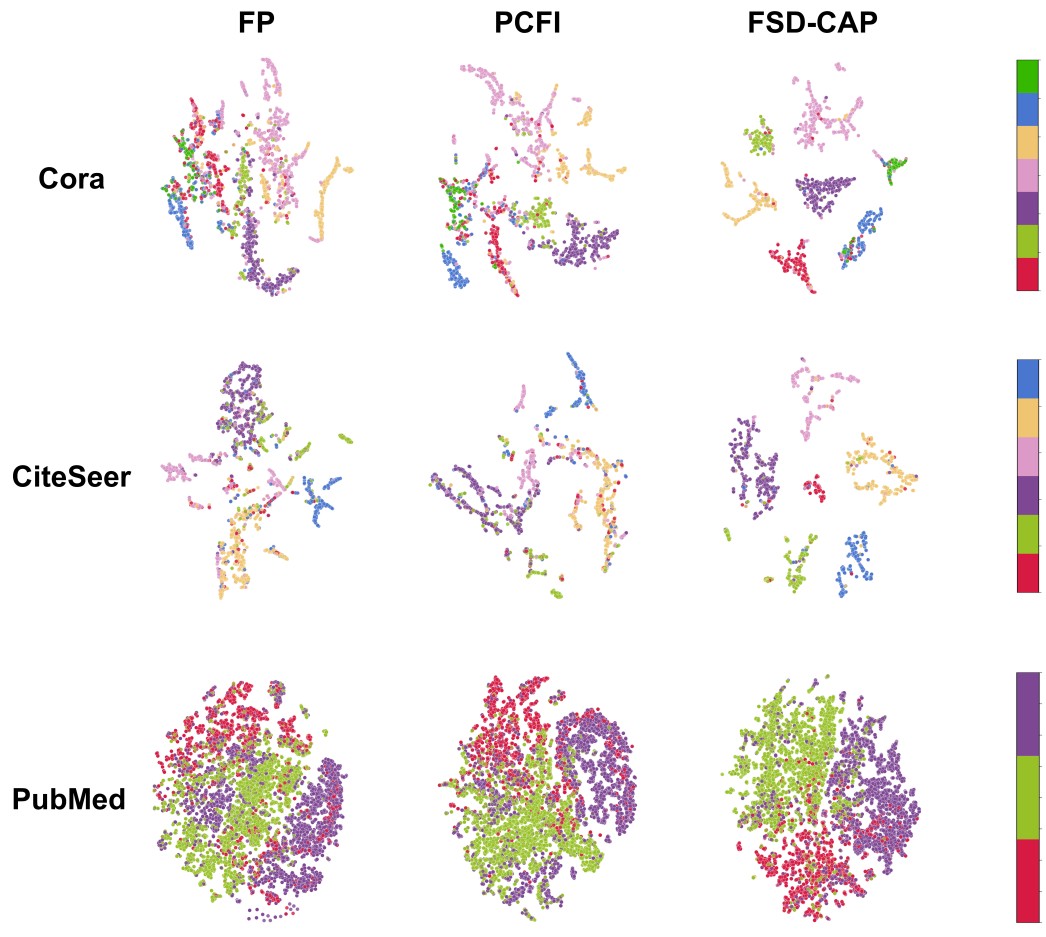

Figure 10: t-SNE visualizations of imputed node features on Cora, CiteSeer, and PubMed under the structural missing setting with a $99.5\%$ missing rate. Nodes are colored by ground-truth class labels.

spent on training the downstream model. The actual feature completion stage, in contrast, is highly efficient, requiring only 0.38s on Cora, 0.76s on CiteSeer, 0.75s on PubMed, 0.57s on Photo, and 1.09s on Computers. Among diffusion-based models, FSD-CAP requires more imputation time than FP and PCFI, yet the cost remains moderate. The extra time comes from the multi-step subgraph expansion, which improves reconstruction under extreme missingness. With only 0.5% of features available, FSD-CAP improves over PCFI by 1.80% to 5.88% on Cora, CiteSeer, PubMed, Photo, and Computers, with an average gain of 3.67%. This yields a favorable trade-off between accuracy and efficiency and is practical for real-world settings with severe feature scarcity.

Table 7: Training time (in seconds) of methods under a structural missing setting with 99.5% missing rate.

|  | Cora | CiteSeer | PubMed | Photo | Computers |
|---|---|---|---|---|---|
| PaGCN | 3.442s | 6.035s | 11.297s | 12.643s | 26.635s |
| GRAFENNE | 78.285s | 30.133s | 225.987s | 160.739s | 223.767s |
| ASD-VAE | 328.785s | 132.977s | OOM | 468.35s | OOM |
| FP | 2.251s | 2.282s | 2.911s | 3.267s | 7.545s |
| PCFI | 2.959s | 2.797s | 3.763s | 3.769s | 8.327s |
| FSD-CAP | 3.276s | 3.103s | 4.022s | 4.329s | 8.992s |

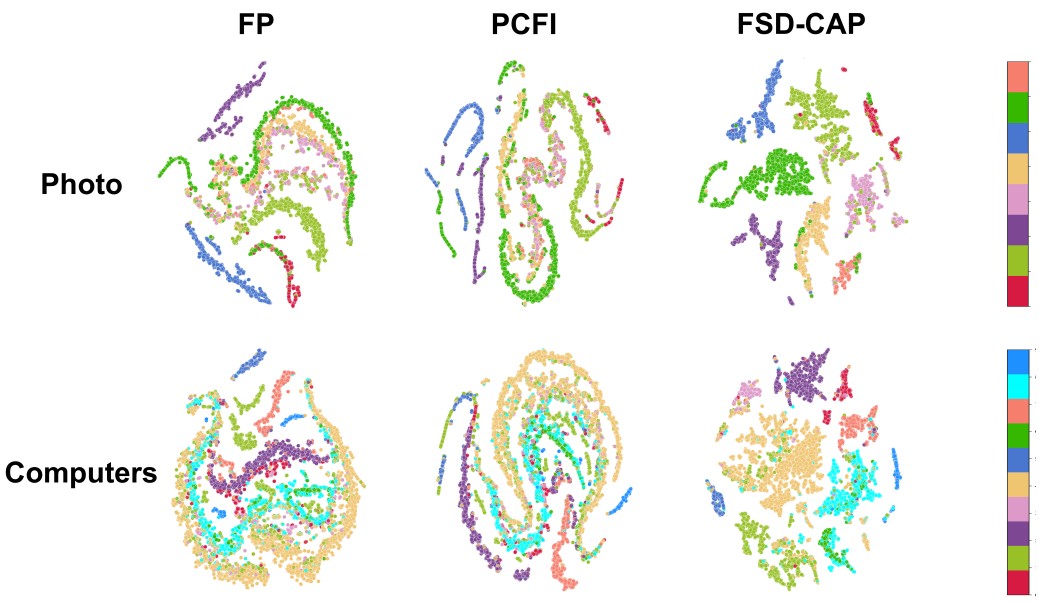

Figure 11: t-SNE visualizations of imputed node features on Photo and Computers under the structural missing setting with a 99.5% missing rate. Nodes are colored by ground-truth class labels.

**Memory complexity.** Table 8 reports the actual memory usage (in MB) of FSD-CAP for both semi-supervised node classification and link prediction under the structural missing setting with a 99.5% missing rate. Results are shown across five datasets, along with the average memory consumption for each task.

Table 8: Memory usage (in MB) of FSD-CAP for semi-supervised node classification and link prediction under the structural missing setting with a 99.5% feature missing rate.

|  | Cora | CiteSeer | PubMed | Photo | Computers | Average |
|---|---|---|---|---|---|---|
| Semi-supervised node classification | 690 | 816 | 3638 | 1014 | 2044 | 1640.4 |
| Link prediction | 658 | 876 | 2560 | 1108 | 2274 | 1495.2 |

### A.2.7 ACCURACY COMPARISON ON LARGE-SCALE DATASETS

We include results on the large-scale OGBN-ArXiv dataset and a subgraph with 50,000 nodes extracted from OGBN-Products. The OGBN-ArXiv graph has 169,343 nodes, 1,166,234 edges, and 128-dimensional features.

We evaluate node classification accuracy under both structural and uniform missing settings. As shown in Table 9, several deep learning models, including GRAFENNE, ITR, and ASD-VAE, run out of memory on two graphs. FSD-CAP attains the best performance among all evaluated methods in both settings. It reaches 69.11% under structural missingness and 70.16% under uniform missingness, outperforming FP and PCFI. In the uniform case, the gap to the full feature model at 72.27% is 2.11 percentage points. These results indicate that the framework scales to large graphs and maintains strong reconstruction quality at extreme sparsity.

### A.2.8 ACCURACY COMPARISON ON HETEROPHILY DATASETS

We evaluate heterophily explicitly. The benchmarks are Cornell, Texas, and Wisconsin from WebKB, and Chameleon from Wikipedia. Their edge homophily ratios are 0.30, 0.11, 0.21, and 0.23, defined as the fraction of edges that connect nodes of the same class.

Table 9: Accuracy comparison on OGBN-Arxiv and OGBN-Products for node classification with 99.5% missing rate.

|  | OGBN-Arxiv | | OGBN-Products | |
| --- | --- | --- | --- | --- |
|  | structural missing | uniform missing | structural missing | uniform missing |
| Full Features | 72.27 ± 0.11 | 72.27 ± 0.11 | 69.04 ± 0.07 | 69.04 ± 0.07 |
| Zero | 56.36 ± 0.63 | 57.70 ± 0.70 | 51.66 ± 1.30 | 58.70 ± 0.70 |
| PaGCN | 47.46 ± 0.51 | 56.92 ± 0.34 | 55.00 ± 0.55 | 56.00 ± 0.56 |
| GRAFENNE | OOM | OOM | OOM | OOM |
| ITR | OOM | OOM | OOM | OOM |
| ASD-VAE | OOM | OOM | OOM | OOM |
| FP | 68.13 ± 0.41 | 68.52 ± 0.18 | 67.53 ± 0.64 | 68.66 ± 0.35 |
| PCFI | 68.59 ± 0.29 | 69.39 ± 0.29 | 68.18 ± 0.30 | 68.39 ± 0.29 |
| **FSD-CAP** | **69.11 ± 0.33** | **70.16 ± 0.27** | **69.03 ± 0.41** | **69.26 ± 0.27** |

We test node classification at a 99.5% missing rate under structural and uniform masks. We compare with diffusion-based completion methods FP and PCFI, and with two heterophily-oriented GNN, Hop GNN and GPR GNN, under the same settings.

As shown in Table 10, when features are extremely sparse, the heterophily GNNs degrade, while completion first helps by reconstructing informative representations. Under structural missingness, Hop GNN averages 44.97%, and FSD-CAP reaches 56.22%. Under uniform missingness, FSD-CAP averages 58.85% and outperforms all baselines.

These findings are consistent with our design. Progressive subgraph diffusion limits early cross-class leakage, and fractional diffusion allows milder mixing on weak homophily. The class-aware refinement then pulls uncertain nodes toward class prototypes and downweights boundary nodes by entropy. Together, the table demonstrates that the framework generalizes well to weak homophily.

Table 10: Accuracy on heterophily graphs for semi-supervised node classification task at 99.5% missing rate. The best results are highlighted in bold.

**Structural Missing**

|  | Chameleon | Texas | Cornell | Wisconsin | Average |
| --- | --- | --- | --- | --- | --- |
| Hop-GNN | 26.65% ± 9.66 | 57.11% ± 6.56 | 47.11% ± 6.56 | 49.02% ± 5.95 | 44.97% |
| GPR-GNN | 21.90% ± 1.75 | 58.42% ± 6.09 | 48.42% ± 6.09 | 46.08% ± 5.49 | 43.70% |
| FP | 39.40% ± 2.97 | 61.07% ± 8.81 | 55.36% ± 12.12 | 55.53% ± 7.20 | 52.84% |
| PCFI | 41.44% ± 7.32 | 64.00% ± 8.11 | 55.17% ± 12.04 | 55.32% ± 7.46 | 53.98% |
| FSD-CAP | **43.23% ± 8.30** | **66.79% ± 6.60** | **57.24% ± 11.14** | **57.63% ± 9.00** | **56.22%** |

**Uniform Missing**

|  | Chameleon | Texas | Cornell | Wisconsin | Average |
| --- | --- | --- | --- | --- | --- |
| Hop-GNN | 25.19% ± 6.75 | 57.11% ± 6.56 | 47.11% ± 6.56 | 49.02% ± 5.95 | 44.61% |
| GPR-GNN | 23.00% ± 1.68 | 58.42% ± 6.09 | 48.42% ± 6.09 | 46.08% ± 5.49 | 43.98% |
| FP | 40.36% ± 4.47 | 61.43% ± 8.57 | 55.86% ± 12.12 | 58.16% ± 4.92 | 53.95% |
| PCFI | 40.54% ± 4.08 | 62.50% ± 9.75 | 55.86% ± 12.12 | 57.37% ± 6.21 | 54.07% |
| FSD-CAP | **47.84% ± 5.21** | **71.07% ± 8.06** | **57.55% ± 11.66** | **58.95% ± 5.79** | **58.85%** |

### A.2.9 ROBUSTNESS ANALYSIS AGAINST DIFFERENT DATA MISSING LEVELS

To evaluate the robustness of FSD-CAP under varying levels of feature missingness, we perform node classification experiments on five datasets under both structural and uniform missing settings. The missing rate is gradually increased from 60% to 99.5%, and results are compared to the full-feature setting to assess how performance degrades with increased sparsity. Table 11 reports the classification accuracy at each missing rate, along with the relative change compared to the fully observed setting.

Table 11: Node classification accuracy (%) of FSD-CAP at different missing rates. Relative performance changes are reported with respect to the fully observed (full-feature) setting.

**Structural Missing**

| Dataset | Full Features | 60% Missing | 70% Missing | 80% Missing | 90% Missing | 95% Missing | 99.5% Missing |
|---|---|---|---|---|---|---|---|
| Cora | 82.72% | 82.75% (+0.03%) | 82.77% (+0.02%) | 82.28% (-0.44%) | 82.45% (-0.27%) | 82.01% (-0.71%) | 80.56% (-0.16%) |
| CiteSeer | 70.00% | 71.29% (+1.29%) | 71.19% (+1.19%) | 71.40% (+1.40%) | 71.97% (+1.97%) | 72.85% (+2.85%) | 71.94% (+1.94%) |
| PubMed | 77.46% | 77.13% (-0.33%) | 76.96% (-0.50%) | 76.89% (-0.57%) | 77.22% (-0.24%) | 77.19% (-0.27%) | 76.98% (-0.48%) |
| Photo | 91.63% | 91.67% (+0.04%) | 91.58% (-0.05%) | 91.34% (-0.29%) | 90.79% (-0.84%) | 90.37% (-1.26%) | 89.18% (-2.45%) |
| Computers | 84.72% | 85.00% (+0.28%) | 84.70% (-0.02%) | 84.13% (-0.59%) | 84.08% (-0.64%) | 84.17% (-0.55%) | 81.64% (-3.08%) |
| Average | 81.31% | 81.57% (+0.26%) | 81.44% (+0.13%) | 81.21% (-0.10%) | 81.30% (-0.01%) | 81.32% (+0.01%) | 80.06% (-1.25%) |

**Uniform Missing**

| Dataset | Full Features | 60% Missing | 70% Missing | 80% Missing | 90% Missing | 95% Missing | 99.5% Missing |
|---|---|---|---|---|---|---|---|
| Cora | 82.72% | 82.84% (+0.12%) | 82.61% (-0.11%) | 82.63% (-0.09%) | 82.61% (-0.11%) | 82.39% (-0.33%) | 81.49% (-1.23%) |
| CiteSeer | 70.00% | 71.34% (+1.34%) | 71.31% (+1.31%) | 72.21% (+2.21%) | 72.26% (+2.26%) | 72.61% (+2.61%) | 73.15% (+3.15%) |
| PubMed | 77.46% | 77.33% (-0.13%) | 77.21% (-0.25%) | 77.38% (-0.08%) | 77.33% (-0.13%) | 76.96% (-0.50%) | 77.46% (-0.00%) |
| Photo | 91.63% | 91.63% (-0.00%) | 91.39% (-0.24%) | 91.02% (-0.61%) | 90.77% (-0.86%) | 90.37% (-1.26%) | 89.40% (-2.23%) |
| Computers | 84.72% | 84.54% (-0.18%) | 84.83% (+0.11%) | 84.30% (-0.42%) | 84.38% (-0.34%) | 84.14% (-0.58%) | 83.57% (-1.15%) |
| Average | 81.31% | 81.54% (+0.23%) | 81.47% (+0.16%) | 81.51% (+0.20%) | 81.47% (+0.16%) | 81.29% (-0.02%) | 81.01% (-0.30%) |

In the structural missing setting, FSD-CAP maintains or exceeds the performance of the full-feature setting when the missing rate is below 95%, demonstrating its ability to handle substantial feature loss. When the missing rate reaches 99.5%, with only 0.5% of features observed, the average accuracy decreases by just 1.25% relative to the fully observed case. In the uniform missing setting, the performance drop is even smaller. At a 99.5% missing rate, the average decline is only 0.3%, confirming the robustness of FSD-CAP under extreme sparsity. On the CiteSeer dataset, FSD-CAP consistently outperforms the full-feature setting across all tested missing rates. Under the 99.5% uniform missing condition, it improves accuracy by 3.15%, indicating that the reconstructed features can be more beneficial for downstream tasks than the original input. These results show that FSD-CAP remains effective and reliable across a wide range of missing levels and patterns.

### A.2.10 ACCURACY COMPARISON WITH SOTA

We further evaluate the performance of FSD-CAP against the current state-of-the-art method (PCFI) on node classification accuracy at different missing rates across five datasets, results are shown in Table 12. FSD-CAP consistently outperforms PCFI across all missing patterns. Notably, as the missing rate increases, the performance gap between FSD-CAP and PCFI becomes more pronounced. Specifically, when the missing rate reaches 99.5%, FSD-CAP achieves an average accuracy improvement of 3.67% across all datasets under the structure missing setting, and 2.24% under the uniform missing setting. On CiteSeer dataset with 99.5% structural missing rate, FSD-CAP achieves a performance gain of 5.88%, demonstrating its robustness to high levels of feature incompleteness compared to PCFI.

Compared to denser graphs such as Photo and Computers, FSD-CAP shows more significant improvements on the sparser datasets (Cora, CiteSeer, and PubMed), with the highest gains observed on the Cora dataset. As revealed by the ablation study in Appendix A.2.1, the second-stage class-aware propagation mechanism plays a critical role in enhancing performance, particularly in sparse graph scenarios. By incorporating class-level feature information, this module effectively enhances inter-class discrimination, thereby improving classification accuracy. In contrast, while the fractional subgraph diffusion stage offers certain improvements over conventional symmetrically normalized diffusion, its overall contribution to performance is relatively modest. On a denser graph with better connectivity, the FSD stage benefits performance by emphasizing local neighborhood influence through a higher fractional exponent $\gamma$, leading to more localized diffusion and improved model behavior.

### A.2.11 SENSITIVITY TO LABEL RATE

To investigate the impact of label availability, we evaluate the performance of FSD-CAP under varying label rates and feature missingness settings. Specifically, we vary the number of labeled nodes per class as $5, 10, 20, 30,$ and $40$, and test under three structural missing rates: 70%, 90%, 99.5%.

Table 12: Node classification accuracy (%) of PCFI and FSD-CAP under different feature missing rates. The performance changes of FSD-CAP relative to PCFI are given in parentheses.

**Structural Missing**

| Dataset | 60% Missing | | 70% Missing | | 80% Missing | | 90% Missing | | 95% Missing | | 99.5% Missing | |
|---|---|---|---|---|---|---|---|---|---|---|---|---|
| | PCFI | FSD-CAP | PCFI | FSD-CAP | PCFI | FSD-CAP | PCFI | FSD-CAP | PCFI | FSD-CAP | PCFI | FSD-CAP |
| Cora | 80.57% | 82.75%(+2.18%) | 80.01% | 82.77%(+2.76%) | 79.36% | 82.28%(+2.92%) | 78.88% | 82.45%(+3.57%) | 77.84% | 82.01%(+4.17%) | 75.36% | 80.56%(+5.20%) |
| CiteSeer | 70.10% | 71.29%(+1.19%) | 69.69% | 71.19%(+1.50%) | 70.21% | 71.40%(+1.19%) | 69.76% | 71.97%(+2.21%) | 69.58% | 72.85%(+3.27%) | 66.06% | 71.94%(+5.88%) |
| PubMed | 76.03% | 77.13%(+1.10%) | 76.09% | 76.96%(+0.87%) | 75.80% | 76.89%(+1.09%) | 75.90% | 77.22%(+1.32%) | 76.07% | 77.19%(+1.12%) | 74.44% | 76.98%(+2.54%) |
| Photo | 91.41% | 91.67%(+0.16%) | 90.88% | 91.58%(+0.70%) | 90.42% | 91.34%(+0.92%) | 89.44% | 90.79%(+1.35%) | 89.09% | 90.37%(+1.28%) | 87.38% | 89.18%(+1.80%) |
| Computers | 84.91% | 85.00%(+0.09%) | 83.63% | 84.70%(+1.07%) | 83.15% | 84.13%(+0.98%) | 82.40% | 84.08%(+1.68%) | 81.90% | 84.17%(+2.27%) | 78.71% | 81.64%(+2.93%) |
| Average | 80.62% | 81.57%(+0.94%) | 80.06% | 81.44%(+1.38%) | 79.79% | 81.21%(+1.42%) | 79.28% | 81.30%(+2.06%) | 78.90% | 81.32%(+2.42%) | 76.39% | 80.06%(+3.67%) |

**Uniform Missing**

| Dataset | 60% Missing | | 70% Missing | | 80% Missing | | 90% Missing | | 95% Missing | | 99.5% Missing | |
|---|---|---|---|---|---|---|---|---|---|---|---|---|
| | PCFI | FSD-CAP | PCFI | FSD-CAP | PCFI | FSD-CAP | PCFI | FSD-CAP | PCFI | FSD-CAP | PCFI | FSD-CAP |
| Cora | 81.01% | 82.84%(+1.83%) | 80.25% | 82.61%(+2.36%) | 80.19% | 82.63%(+2.44%) | 79.55% | 82.61%(+3.06%) | 78.59% | 82.39%(+3.80%) | 78.55% | 81.49%(+2.94%) |
| CiteSeer | 71.10% | 71.34%(+0.24%) | 69.97% | 71.31%(+1.34%) | 70.13% | 72.21%(+2.08%) | 69.92% | 72.26%(+2.34%) | 68.85% | 72.61%(+3.76%) | 69.11% | 73.15%(+4.04%) |
| PubMed | 76.29% | 77.33%(+1.04%) | 75.90% | 77.21%(+1.31%) | 76.18% | 77.38%(+1.20%) | 76.58% | 77.33%(+0.75%) | 76.26% | 76.96%(+0.70%) | 76.01% | 77.46%(+1.45%) |
| Photo | 90.88% | 91.63%(+0.75%) | 90.97% | 91.39%(+0.42%) | 90.42% | 91.02%(+0.60%) | 89.73% | 90.77%(+1.04%) | 89.08% | 90.37%(+1.29%) | 88.55% | 89.40%(+0.85%) |
| Computers | 83.62% | 84.54%(+0.92%) | 83.40% | 84.83%(+1.43%) | 83.39% | 84.30%(+0.91%) | 83.14% | 84.38%(+1.24%) | 82.42% | 84.14%(+1.72%) | 81.64% | 83.57%(+1.93%) |
| Average | 80.58% | 81.54%(+0.96%) | 80.10% | 81.47%(+1.37%) | 80.06% | 81.51%(+1.45%) | 79.78% | 81.47%(+1.69%) | 79.04% | 81.29%(+2.25%) | 78.77% | 81.01%(+2.24%) |

Table 13: Accuracy(%) of FSD-CAP on Cora for node classification task under varying feature missing rates and label rates.

| | 5 | 10 | 20 | 30 | 40 |
|---|---|---|---|---|---|
| 70% missing | 76.41 ± 3.21 | 78.42 ± 1.29 | 80.49 ± 1.86 | 81.33 ± 1.97 | 81.39 ± 1.86 |
| 90% missing | 75.59 ± 2.98 | 78.25 ± 1.73 | 80.27 ± 1.16 | 81.43 ± 1.54 | 81.27 ± 1.16 |
| 99.5% missing | 74.49 ± 2.94 | 78.07 ± 1.87 | 79.55 ± 1.57 | 79.89 ± 2.01 | 79.95 ± 1.65 |

Table 14: Accuracy(%) of FSD-CAP on Photo for node classification task under varying feature missing rates and label rates.

| | 5 | 10 | 20 | 30 | 40 |
|---|---|---|---|---|---|
| 70% missing | 86.84 ± 1.98 | 88.18 ± 1.71 | 91.35 ± 0.56 | 91.57 ± 0.35 | 91.43 ± 0.48 |
| 90% missing | 86.39 ± 1.98 | 87.01 ± 2.08 | 90.60 ± 0.54 | 91.17 ± 0.44 | 90.80 ± 0.52 |
| 99.5% missing | 84.52 ± 2.32 | 84.93 ± 2.19 | 88.98 ± 1.01 | 89.42 ± 0.92 | 88.88 ± 0.97 |

Table 13 and Table 14 report node classification accuracy on Cora and Photo under the structural missing setting, respectively. The results show that the performance of FSD-CAP consistently improves as the number of labeled nodes per class increases from 5 to 30, and the gains begin to saturate when reaching 30–40 labels per class, indicating that the model has effectively leveraged the available supervision and approached convergence under the current configuration.

We further evaluate FSD-CAP in an extreme low-label setting (5 labels per class), combined with 99.5% feature missingness. We compare FSD-CAP against several recent state-of-the-art baselines in this challenging setting. As shown in Table 15, while all methods suffer performance degradation due to the scarcity of supervision signals, FSD-CAP still significantly outperforms all competitors. This demonstrates the effectiveness of the class-level prior construction, which enables more robust propagation even when labeled data are extremely limited.

Table 15: Classification accuracy (%) under 99.5% structural missingness and 5 labeled nodes per class. The best results are highlighted in bold.

| | Cora | CiteSeer | PubMed | Photo | Computers | Average |
|---|---|---|---|---|---|---|
| FP | 74.45 ± 3.16 | 67.05 ± 2.64 | 64.71 ± 9.10 | 82.01 ± 3.06 | 71.07 ± 4.56 | 71.86 |
| PCFI | 67.76 ± 3.46 | 58.68 ± 4.55 | 66.91 ± 6.46 | 81.26 ± 2.02 | 73.30 ± 2.09 | 69.58 |
| FSD-CAP | **74.49 ± 2.94** | **68.26 ± 2.59** | **68.79 ± 3.13** | **84.52 ± 2.32** | **75.71 ± 1.86** | **74.35** |

### A.3 Implementation and Hyperparameters

#### A.3.1 Datasets

We evaluate FSD-CAP on five benchmark datasets: three citation networks (**Cora**, **CiteSeer**, and **PubMed**) and two Amazon co-purchase networks (**Photo** and **Computers**).

- In the citation networks, nodes represent academic papers and edges indicate citation links. Node features are bag-of-words representations of paper abstracts, and each node is labeled according to its research topic.
- In the Amazon co-purchase networks, nodes correspond to products and edges connect items frequently purchased together. Node features are derived from bag-of-words encodings of product reviews, and labels reflect product categories.

All datasets are publicly available through the MIT-licensed PyTorch Geometric library. We evaluate on the largest connected component of each graph. Dataset statistics are provided in Table 16, using the setup in FP and PCFI.

Table 16: Dataset statistics and data splits used for semi-supervised node classification.

| Dataset | Nodes | Edges | Attributes | Classes | Train/Valid/Test Nodes |
|---|---|---|---|---|---|
| Cora | 2,485 | 5,069 | 1,433 | 7 | 140/1,360/985 |
| CiteSeer | 2,120 | 3,679 | 3,703 | 6 | 120/1,380/620 |
| PubMed | 19,717 | 44,324 | 500 | 3 | 60/1,440/18,217 |
| Photo | 7,487 | 119,043 | 745 | 8 | 160/1,340/5,987 |
| Computers | 13,381 | 245,778 | 767 | 10 | 200/1,300/11,881 |

#### A.3.2 Data Split

For node classification, we follow a two-step data split. First, 1500 nodes are randomly selected as the development set, with the remainder used for testing. Within the development set, 20 nodes per class are randomly sampled for training, and the rest form the validation set, following the setup in FP and PCFI. Table 16 summarizes the data split for each dataset.

For link prediction, we adopt the edge split protocol from PCFI, allocating $85\%$ of edges for training, $5\%$ for validation, and $10\%$ for testing. To ensure robust evaluation, all experiments are repeated with 10 random seeds. For each seed, we generate independent data splits and feature mask matrices $M$ to simulate missing attributes. The final results are reported as the mean and standard deviation over these 10 runs.

The feature mask matrix $M$ is initialized as an all-ones matrix with the same shape as the original feature matrix. Under uniform missing, $m\%$ of the entries are randomly set to zero. Under structural missing, $m\%$ of the rows are randomly selected and set entirely to zero, simulating nodes with fully missing features.

#### A.3.3 Baseline Details

We compare FSD-CAP with the following methods:

**i. Zero.** A standard GCN trained on the feature matrix with missing values replaced by zeros. We use the implementation from the PyTorch Geometric library (MIT License).

**ii. PaGCN.** PaGCN applies partial graph convolution over observed features without explicitly modeling missingness. We use the authors' MIT-licensed code.[2]

**iii. ITR.** ITR performs adaptive imputation through a two-stage process: initialization from graph structure followed by refinement using affinity updates. We use the Apache-2.0 licensed implementation.[3]

---

[2] https://github.com/yaya1015/PaGCN
[3] https://github.com/WxTu/ITR

**iv. ASD-VAE.** ASD-VAE learns a shared latent space by maximizing the joint likelihood of attribute and structure views, then reconstructs features via decoupled decoding. We use the authors' publicly released code.[4]

**v. FP.** FP reconstructs features through iterative propagation using the normalized adjacency matrix. We use the Apache-2.0 licensed code.[5]

**vi. PCFI.** PCFI imputes missing values by performing inter-node and inter-channel diffusion weighted by pseudo-confidence. We use the official Apache-2.0 licensed implementation.[6]

**vii. GRAFENNE.** GRAFENNE uses a three-phase message-passing framework on an allotropically transformed graph to learn from streaming features. We use the authors' released code.[7]

### A.3.4 EVALUATION METRICS

We evaluate FSD-CAP on two standard graph learning tasks: semi-supervised node classification and link prediction.

**Semi-supervised Node Classification.** Given a graph with partially labeled nodes, the objective is to predict labels for the unlabeled nodes. Performance is measured by classification accuracy, defined as the proportion of correctly predicted labels across all nodes. Higher accuracy indicates better generalization in the semi-supervised setting.

**Link Prediction.** This task involves predicting missing edges between node pairs using learned feature representations. We use AUC (area under the ROC curve) and AP (average precision) as evaluation metrics. AUC quantifies the model's ability to rank existing edges above non-edges, while AP measures the area under the precision-recall curve, reflecting the balance between precision and recall.

### A.3.5 EXPERIMENTAL SETTINGS

**Semi-supervised Node Classification.** All experiments use a consistent data split strategy with 10 random seeds per dataset. All models, except ASD-VAE and PaGCN, are implemented using a three-layer GCN trained with the Adam optimizer. ASD-VAE uses the Katz-GCN architecture as proposed in its original work, and PaGCN employs a modified GCN designed for incomplete inputs. For methods with reported hyperparameters (ITR, GRAFENNE, ASD-VAE), we follow the original configurations. For the remaining models, the learning rate is selected from $\{0.1, 0.01, 0.005, 0.001, 0.0005\}$, and dropout from $\{0.0, 0.25, 0.5\}$, based on validation performance. All experiments are run on a 40-core Intel Xeon Silver 4210R CPU with 4 NVIDIA RTX 4090 GPUs (24GB each).

All methods are evaluated under both structural and uniform missing settings across varying missing rates $mr$, except ITR, which is applicable only to structural missing where node features are either fully observed or fully missing.

**Link Prediction.** We adopt the same 10-split strategy across all datasets. All models are implemented using a two-layer Graph Autoencoder and trained with Adam. Hyperparameters are tuned on the validation set, with the same learning rate and dropout search space used in the node classification task.

### A.3.6 FSD-CAP IMPLEMENTATION

For semi-supervised node classification, we set the GCN learning rate to $0.01$ with dropout $0.5$ on Cora, CiteSeer, and Computers. On PubMed and Photo, the learning rate is $0.005$, with dropout values of $0.5$ and $0.25$, respectively. The pre-trained GCN used for pseudo-label generation adopts the same settings. Following PCFI, the number of propagation steps $K$ is fixed at $100$, sufficient for convergence.

---

[4]`https://github.com/jiangxinke/ASD-VAE`
[5]`https://github.com/twitter-research/feature-propagation`
[6]`https://github.com/daehoum1/pcfi`
[7]`https://github.com/data-iitd/Grafenne`

We tune the hyperparameters $\gamma$ (fractional diffusion exponent), $\lambda$ (feature retention coefficient), and $T$ (temperature) using grid search. For sparse datasets (Cora, CiteSeer, PubMed), $\gamma$ is selected from $[0.6, 1.6]$; for dense datasets (Photo, Computers), from $[2.0, 4.0]$, both with a step size of $0.2$. $\lambda$ is searched in $[0, 1]$ with step $0.1$. $T$ is selected from $\{1, 5, 25, 100, 250, 1000\}$.

In the link prediction task, the GCN classifier uses the same configuration as above. For the downstream Graph Autoencoder, the dropout rate is fixed at $0.5$. Learning rates are set to $0.005$ for Cora and CiteSeer, $0.1$ for PubMed, and $0.0005$ for Photo and Computers. Hyperparameter tuning follows the same protocol: $\lambda \in [0, 1]$ with step $0.1$, $T \in \{0.001, 0.01, 0.1, 1, 5, 10, 25, 50\}$. For $\gamma$, we use $[4.0, 6.0]$ on dense datasets and $[0.6, 1.6]$ on sparse datasets, both with step $0.2$.

We report the selected hyperparameter configurations for semi-supervised node classification and link prediction under a $99.5\%$ feature missing rate in Table 17 and Table 18, respectively. Additional parameter sensitivity analysis is presented in Appendix A.2.2.

Table 17: Hyperparameter configurations for FSD-CAP on semi-supervised node classification task.

|  | Parameter | Cora | CiteSeer | PubMed | Photo | Computers |
|---|---|---|---|---|---|---|
| Structural Missing | $\gamma$ | 1.2 | 1.2 | 1.6 | 2.8 | 3.8 |
|  | $\lambda$ | 0.2 | 0.9 | 0.6 | 0.3 | 0.4 |
|  | $T$ | 5 | 250 | 5 | 25 | 100 |
| Uniform Missing | $\gamma$ | 1.2 | 1.2 | 1.2 | 2.8 | 4.0 |
|  | $\lambda$ | 0.2 | 0.7 | 0.1 | 0.0 | 0.3 |
|  | $T$ | 250 | 250 | 5 | 25 | 100 |

Table 18: Hyperparameter configurations for FSD-CAP on link prediction task.

|  | Parameter | Cora | CiteSeer | PubMed | Photo | Computers |
|---|---|---|---|---|---|---|
| Structural Missing | $\gamma$ | 1.4 | 1.4 | 1.4 | 4.2 | 5.0 |
|  | $\lambda$ | 0.2 | 0.3 | 0.0 | 0.8 | 0.0 |
|  | $T$ | 5 | 25 | 10 | 0.001 | 0.001 |
| Uniform Missing | $\gamma$ | 1.2 | 1.6 | 1.6 | 4.4 | 5.6 |
|  | $\lambda$ | 0.0 | 0.0 | 0.0 | 0.0 | 0.0 |
|  | $T$ | 5 | 25 | 25 | 0.01 | 0.001 |

The implementation of FSD-CAP (Apache-2.0 licensed) will be made publicly available upon publication.

### A.3.7 ADDITIONAL DISCUSSION

**Limitations.** This work focuses solely on missing node attributes and does not address missing or uncertain edges. In many practical settings, both node features and graph topology may be incomplete. Extending FSD-CAP to handle joint feature and structure imputation remains an open and important direction for future research.

**Societal Impacts.** FSD-CAP improves the robustness of GNNs under high feature-missing rates and may benefit applications in domains such as social networks and recommender systems. However, as with any imputation technique, there is a risk of misuse, particularly in inferring sensitive or private attributes from partial data. We encourage responsible use, including proper access controls and ethical oversight, especially when applying the model to contexts involving personal or sensitive information.

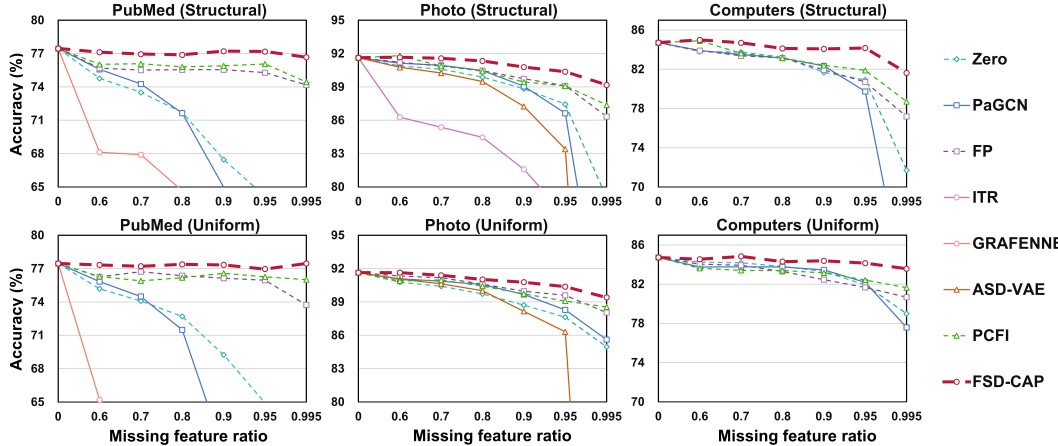

Figure 12: Node classification accuracy (%) on PubMed, Photo and Computers for $mr \in \{0.6, 0.7, 0.8, 0.9, 0.95, 0.995\}$. The top row shows structural missing; the bottom row shows uniform missing. Methods that are inapplicable or result in out-of-memory errors are omitted.

## A.4 SUPPLEMENTARY FIGURES AND TABLES

### A.4.1 ACCURACY ON PUBMED, PHOTO AND COMPUTERS UNDER VARYING MISSING RATES.

This subsection presents the classification accuracy results for the PubMed, Photo and Computers datasets under structural and uniform missing scenarios, with missing rates (mr) ranging from $0.6$ to $0.995$. The corresponding figure is shown in Figures 12.

## A.5 DECLARATION OF LLM USAGE

The use of large language models (LLMs) is only for editing purposes such as grammar, spelling, and formatting checks, and does not influence the core methodology, scientific rigor, or originality of this research. LLMs are not involved in the design, analysis, or interpretation of the study.

## A.6 PSEUDOCODE OF FRACTIONAL SUBGRAPH DIFFUSION WITH CLASS-AWARE PROPAGATION (FSD-CAP)

### A.6.1 PSEUDOCODE OF FRACTIONAL SUBGRAPH DIFFUSION (ALGORITHM 1)

### A.6.2 PSEUDOCODE OF CLASS-AWARE PROPAGATION (ALGORITHM 2)

---

**Algorithm 1** FSD algorithm

---

**Input:** Graph $\mathcal{G} = \{X, A\}$, Known/unknown node sets $\mathcal{V}_{+/-}^{\ell}$ for each channel $\ell$, Binary mask matrix $M$, True labels $y$, Number of channels $F$ and node $N$, Hyper-parameters $(\gamma, \text{K}, \lambda)$

1: **for** channel $\ell = 1, \ldots, F$ **do**
2:     **Initialize** sub-graph $\mathcal{G}^{(0)}$:
3:         Keep starting $m = 0$   *(Initialize the number of sub-layers)*
4:         Read subgraphs $A^{(m)} =$k-hopSubgraph$(\mathcal{V}_+^{\ell}, A, m)$   *(Extract A of m- hop subgraph)*
5:     **while** $A^{(m)} \neq A$ **do**
6:         Normalize $\mathbf{A}^{(m)} = D^{-1/2} A^{(m)} D^{-1/2}$   *(Symmetric normalized adjacency matrix)*
7:         Fractional weighted matrix $\mathbf{A}_{ij}^{\gamma;m} = (\mathbf{A}_{ij}^{(m)})^{\gamma} / \left( \sum_{k=1}^{N} (\mathbf{A}_{ik}^{(m)})^{\gamma} \right)$
8:         **for** $t = 1$ to $K$ **do**
9:             $\mathbf{x}^{(m)}(t) = \mathbf{x}^{(m)}(0) \odot M + (\mathbf{A}^{\gamma,m} \mathbf{x}^{(m)}(t-1) + \lambda \mathbf{x}^{(m-1)}(K)) \odot (1 - M)$
10:         **end for**
11:         $m \leftarrow m + 1$   *(Enter the next level of subgraph)*
12:         Extended subgraphs $A^{(m)} =$k-hopSubgraph$(\mathcal{V}_+^{\ell}, A, m)$
13:     **end while**
14:     $\mathbf{x}^{\ell} = \mathbf{x}^{(m)}(K)$   *(Update final features of the current channel)*
15: **end for**
16: Stack $\{\mathbf{x}^{\ell}\}_{\ell=1}^{F}$ to form pre-imputed $\tilde{X}$
17: **return** Pre-imputed feature matrix $\tilde{X}$

---

**Algorithm 2** CAP algorithm

---

**Input:** Graph $\mathcal{G} = \{\tilde{X}, A\}$, Known/unknown node sets $\mathcal{V}_{+/-}^{\ell}$ for each channel $\ell$, Binary mask matrix $M$, True labels $y$, Number of node $N$ and classes $C$, Hyper-parameters $T$

1: Get pseudo-labels $\tilde{y}$ and predicted class probability $\hat{y}$ using GNN with $(A, \tilde{X}, y)$ and temperature $T$
2: **for** $i = 1$ to $N$ **do**
3:     **for** $c = 1$ to $C$ **do**
4:         $P_i(c) = \left(1/|\hat{\mathcal{N}}_i|\right) \sum_{j \in \mathcal{N}_i} \mathbb{1}_{(\tilde{y}_j = c)}$
5:     **end for**
6:     information entropy $S_i = - \left(1/\log\left(|\hat{\mathcal{N}}_i|\right)\right) \sum_{c \in C} P_i(c) \cdot \log\left(P_i(c)\right)$
7: **end for**
8: **for** $c = 1$ to $C$ **do**
9:     compute class-specific feature: $x_{(c)}^{\star} = \left(\sum_{\tilde{y}_i = c}(1 - S_i) \cdot x_i\right) / \left(\sum_{\tilde{y}_i = c}(1 - S_i)\right)$
10:     Define virtual class node $v^{(c)}$ and class subset $\mathcal{V}^{(c)} = \{v^{(c)}\} \cup \mathcal{V}_-^{(c)}$
11:     $X_-^{(c)} = $ extract_matrix$(\tilde{X}, \mathcal{V}_-^{(c)})$   *($X_-^{(c)}$ contains imputed features of nodes in $\mathcal{V}_-^{(c)}$)*
12:     Feature Matrix for $\mathcal{V}^{(c)}$: $X^{(c)} = [X_-^{(c)} \;\; x_{(c)}^{\star}]^T$   *(row-wise concatenation)*
13:     Initialize $\mathbf{W}^{(c)}$ as zero matrix of size $|\mathcal{V}^{(c)}| \times |\mathcal{V}^{(c)}|$
14:     **for** each node $i \in \mathcal{V}^{(c)}$ **do**
15:         $\mathbf{W}_{ii}^{(c)} = \hat{y}_{i,c}$   *(self-loop weight: confidence in class c)*
16:         $\mathbf{W}_{i,|\mathcal{V}^{(c)}|}^{(c)} = 1 - \hat{y}_{i,c}$   *(edge from node $v_i$ to class node $v^{(c)}$)*
17:     **end for**
18:     Class Propagation $\hat{X}^{(c)} = \mathbf{W}^{(c)} X^{(c)}$
19:     Class feature matrix $X_c = $ extract_matrix$(\hat{X}^{(c)}, \mathcal{V}_-^{(c)})$
20:     $\hat{X} = $ combine_matrix$(\tilde{X}, X_c, M)$   *(restore known features and update refined features)*
21: **end for**
22: **return** Imputed feature matrix $\hat{X}$

---

