# OpenReview forum: "FSD-CAP: Fractional Subgraph Diffusion with Class-Aware Propagation for Graph Feature Imputation"
_ICLR.cc/2026/Conference — ICLR 2026 Poster_

### Official Review · Reviewer_fH1t · 2025-10-29

**Soundness:** 3
**Presentation:** 2
**Contribution:** 2
**Rating:** 4
**Confidence:** 1

**Summary:**

This paper addresses the challenge of graph feature imputation under extreme missing rates—a limitation of existing methods that fail to produce reliable estimates or propagate errors in sparse graphs. The authors propose FSD-CAP, a two-stage diffusion-based framework designed to enhance imputation robustness and adaptability:

**Strengths:**

1. FSD-CAP combines three components (fractional diffusion, progressive subgraphs, class-aware refinement) in a cohesive framework, avoiding the "black-box" nature of many deep learning-based imputers.
2. The method’s two-stage pipeline is visualized and described in plain terms, making it accessible to researchers new to graph diffusion. Appendices provide implementation code details, hyperparameter ranges, and extra experiments (e.g., accuracy vs. distance from observed nodes).

**Weaknesses:**

1.  The paper only addresses missing node attributes, ignoring missing or noisy edges—a common real-world scenario (e.g., incomplete social network connections). Extending FSD-CAP to joint feature-structure imputation is mentioned as future work but is a key limitation for broader applicability.
2.  Experiments focus on standard benchmarks (citation networks, product co-purchase graphs) but lack a real-domain application (e.g., imputing missing gene features in biological networks, user attributes in recommendation systems). A case study would strengthen the paper’s practical impact.

**Questions:**

1. How would FSD-CAP need to be modified to handle both missing node features and missing edges? For example, could the subgraph expansion stage incorporate edge confidence scores to avoid propagating information through uncertain edges?
2. Most experiments use static graphs. Would FSD-CAP’s progressive subgraph expansion adapt to dynamic graphs (e.g., evolving social networks with new nodes/edges), or would frequent subgraph recomputation be computationally prohibitive?

---

> ### Author Response · Authors · 2025-11-23
> **Response to Reviewer fH1t --- PartⅠ**
>
> **Weaknesses:**
>
> **W1:** Graph neural networks (GNNs) have achieved remarkable success in various graph learning tasks, yet they typically assume that node attributes are fully observed. In practice, this assumption may hardly hold in the real world since graph attributes are possibly incomplete or missing due to a variety of subjective or objective factors. The performance of GCNs degrades sharply when the feature missing rate exceeds 70\% (Figure 2 in our manuscript). This highlights a critical need to enhance the robustness of GNNs when operating on graphs with incomplete attribute information.
>
> Accordingly, our work focuses on addressing this fundamental bottleneck, i.e., how to reconstruct high-quality node features from extremely sparse observations, thereby providing reliable inputs for downstream GNNs. We believe this is a necessary first step toward building robust graph learning systems under real-world data scarcity.
>
> We agree with the reviewer that missing edges and missing features often co-occur in practical scenarios. Handling such complex missingness would require integrating feature imputation with edge reconstruction, e.g., by cascading or jointly optimizing a diffusion-based feature completion module with a structure learning component. However, such joint optimization introduces nontrivial challenges, including the coupling between fractional diffusion operators and graph topology refinement. This direction is beyond the scope of the current work but is indeed a key focus of our planned future research.
>
> **W2:** Thank you for raising such insightful questions. In our previous experiments, we conducted tests on a series of well-established homogeneous datasets, such as Cora, CiteSeer, PubMed, Computer, and Amazon Photo. Additionally, we also carried out experiments on diverse heterophily datasets, including Chameleon, Texas, and Cornell. These experiments hold certain practical significance as they assist us in evaluating the model's performance across multiple scenarios and domains.
>
> From the perspective of enhancing the real-world applicability and broader influence of our research, your viewpoint regarding the lack of real-domain applications in our current experiments is indeed well-grounded and highly meaningful. In our future research, we aspire to apply our model to more specific real-world cases, such as imputing missing gene features in biological networks or inputting user attributes in recommendation systems. We plan to investigate the issues associated with these practical applications. For instance, how to handle the complex and noisy data in real-world biological networks or how to strike a balance between accuracy and efficiency when dealing with large-scale user data in recommendation systems.

---

> ### Author Response · Authors · 2025-11-23
> **Response to Reviewer fH1t --- PartⅡ**
>
> **Questions:**
>
> **Q1:** Thank you for your valuable suggestions. Regarding the technical improvement directions for the FSD-CAP model in handling scenarios with both node feature and edge missingness, we have conducted in-depth theoretical analysis and explored the feasibility of introducing an Edge Confidence Score mechanism based on your advice.
>
> In scenarios with dual missingness, traditional diffusion models (such as FP and PCFI) may suffer from ineffectiveness in information propagation paths due to edge missingness, which in turn leads to the accumulation of errors in feature reconstruction. This is because edge missingness can disrupt the original propagation priority of high-confidence nodes, whereas the Edge Confidence Score can alleviate this issue by quantifying edge reliability and providing dynamic weights for information propagation. Considering the dual missingness scenario, the subgraph expansion stage of FSD-CAP should be improved from solely relying on the topological structure of the graph to a hybrid consideration of both topological structure and edge confidence scores, with relevant hybrid schemes such as weighted propagation mechanisms and dynamic threshold filtering taken into account.
>
> We sincerely appreciate the reviewer's suggestions, which have provided significant inspiration for addressing the dual missingness problem. Subsequent research will focus on conducting experimental validation around the aforementioned directions and refining the theoretical analysis.
>
>
> **Q2:** Thank you for this insightful question. FSD-CAP consists of two stages. The first (progressive subgraph diffusion) performs iterative diffusion over a small number of expanded subgraphs. The second (class-aware refinement) is a single propagation on a compact class-level graph and adds negligible overhead.
>
> In static settings, the method is already highly efficient. 100 iterations are sufficient for convergence across all datasets, which balances accuracy and efficiency. Only 4 to 7 expansion rounds are needed to cover the graph effectively. For example, on Cora, one fractional-diffusion pass (100 iterations) over a subgraph takes ~60 ms, with six expansion rounds take ~380 ms in total. The subsequent CAP stage adds just ~6 ms, resulting in an imputation time of less than 0.4 seconds.
>
> The layer-wise and localized design of the progressive diffusion stage naturally supports adaptation to dynamic graphs. When edges or nodes change, we do not need to recompute the entire diffusion from scratch. Instead, only the affected rows of the fractional diffusion matrix $A^\gamma$ are updated and renormalized. Diffusion is re-run locally on the impacted frontier (i.e., neighborhoods around changed edges/nodes) with a warm start from the previous layer’s features, enabling rapid convergence. Because the update cost scales with the number of changed edges rather than the full graph size, FSD-CAP remains computationally feasible even under frequent structural changes.

---

### Official Review · Reviewer_vLJV · 2025-10-29

**Soundness:** 2
**Presentation:** 3
**Contribution:** 3
**Rating:** 4
**Confidence:** 4

**Summary:**

This paper introduces FSD-CAP, a two-stage framework for imputing missing node features, especially in the cases with high sparsity of node metadata. The framework performs an initial imputation based on "fractional diffusion operator" to adjust propagation sharpness based on local structure, with "progressive subgraph diffusion" mechanism that expands layer-by-layer from observed nodes, localizing diffusion and limiting error propagation. Then, Class-Aware Propagation (CAP) refines the imputed features by applying a standard GCN to the FSD-imputed features to generate synthetic class labels. CAP then creates one vector for each class, and this class-specific vectors are formed based on the nodes inside each class but not those in the boundaries (though the "credibility" weight calculated by the faction of neighbors sharing the same label with each focal node in a class.) By down-weighting the features from the boundary nodes and imputing these class-specific vectors into the nodes in its class, this class-specific vectors serve as counteractive mechanism against over-smoothing. The effectiveness is demonstrated using the node classification and link prediction benchmarks, confirming that the proposed method is robust against the missing features of nodes.

**Strengths:**

The authors introduced a practical solution to balance imputation and over-smoothing though the fractional diffusion operator and CAP as two counteracting mechanisms. The authors also demonstrated the effectiveness in node classification and link prediction problems, which demonstrate the generalizability of the proposed solution for the missing feature problem.

**Weaknesses:**

W1
The paper would be clearer if it explicitly stated the key assumptions underlying the proposed solution. In essence, the methods succeed only when three conditions hold: (1) the node features are informative for the task at hand; (2) the features are homophilic (adjacent nodes tend to share similar values); and (3) the graph exhibits community structure in which each community is dominated by nodes with the same feature label. It is important to note that node metadata do not always reflect the network topology, and not every metadata type is useful for a given application (see [1]). If condition (2) fails (i.e., neighboring nodes are likely to have different features), the diffusion process will incorrectly impute missing values. Moreover, the CAP framework relies on the premise that node features and community structure are strongly coupled. The fractional diffusion operator is built from the normalized adjacency matrix, and its mixing behavior is largely dictated by the community structure (as captured by the Fiedler vector). When node features are not correlated with the community structure, the diffusion does not remain localized; consequently, many nodes have similar feature values, and CAP loses its ability to distinguish clusters within the network.

I'm overall positive about the ideas from the authors but suggest them to make it very explicit about the key premises the method is based on, and clarify when the proposed solution works and do not work.

[1] Peel, Leto, Daniel B. Larremore, and Aaron Clauset. "The ground truth about metadata and community detection in networks." Science advances 3.5 (2017): e1602548.

W2
I appreciate the authors to include GCN as a baseline, with missing features imputed with zeros. The paper could be benefit from providing other common heuristics and the comparisons against more recent convolution based graph neural networks. For example, another common imputation method is to fill the missing feature with a global mean. Alongside is the k-nearest neighbor methods. These are pretty common imputation methods that do not require graph structure (and thus are less prone to the violation of the assumptions discussed above). Also GCN (Kipf & Welling, 2017) is already primitive, and the GNN field has advanced significantly since 2017. I suggest the authors to include advanced and commonly used models (not necessarily the state-of-the-art models, such as GAT) to demonstrate the broader applicability.

**Questions:**

Q1: What are the underlying premise for the proposed solution?

Q2: What are the performance gain of the proposed imputation methods against traditional but well-accepted generic imputation methods (i.e., global mean and k-nn based method)?

---

> ### Author Response · Authors · 2025-11-23
> **Response to Reviewer vLJV --- PartⅠ**
>
> **Weakness 1 & Question 1:**
>
> We sincerely thank you for your insightful comments. The proposed model builds upon feature propagation by performing progressive diffusion-based completion. Like other diffusion-based imputation methods (e.g., FP, PCFI), our approach implicitly assumes a degree of feature homophily, that is, connected nodes tend to have similar attributes. This aligns with the common design principle in many GNNs, which model homophily by propagating and aggregating features across local neighborhoods through various message-passing mechanisms [1].
>
> To enhance feature discriminability after diffusion, the CAP (Class-Aware Propagation) stage constructs several class-level graphs, rather than operating on the original graph, and propagates synthesized class-anchor features across these semantic structures. This can be viewed as an implicit clustering process, which leverages the assumption that nodes within the same class exhibit certain feature similarity, even if they are not directly connected in the original graph.
>
> As the reviewer points out, when these assumptions are violated, particularly in heterophilic graphs, where neighboring nodes often belong to different classes, the effectiveness of smoothness-based diffusion may degrade. To evaluate robustness under such non-ideal conditions, we conduct experiments on four standard heterophilic benchmark (Cornell, Texas, Wisconsin, and Chameleon) with edge homophily ratios ranging from 0.11 to 0.30.
>
> We test node classification at a 99.5\% missing rate under both structural and uniform settings. We compare with diffusion-based completion methods FP, PCFI, two recent state-of-the-art diffusion-based methods FISF [2] and GOODIE [3], as well as two GNNs designed for heterophily, namely, Hop-GNN [4] and GPR-GNN [5], under the same missingness.
>
> The table below reports accuracies on the four heterophily graphs over 10 runs. When features are extremely sparse, the heterophily GNNs degrade, while completion first helps by reconstructing informative representations. FSD-CAP is superior across all datasets. Under structural missingness, Hop-GNN averages 44.97\% and FSD-CAP reaches 56.22\%. Compared with PCFI, FSD-CAP improves accuracy by 1.79\% on Chameleon, 2.79\% on Texas, 2.07\% on Cornell, and 2.31\% on Wisconsin.
>
> **Table 1:** Accuracy(\%) on heterophily graphs for node classification task at 99.5\% missing rate.
> |Method|Chameleon|Texas|Cornell|Wisconsin|Average|
> |------------|----------------|-----------------|-----------------|-----------------|---------|
> |Hop-GNN| 26.65 ± 9.66|57.11 ± 6.56|47.11 ± 6.56|49.02 ± 5.95|44.97|
> |GPR-GNN| 21.90 ± 1.75|58.42 ± 6.09|48.42 ± 6.09|46.08 ± 5.49|43.70|
> |FP|39.40 ± 2.97| 61.07 ± 8.81| 55.36 ± 12.12| 55.53 ± 7.20|52.84|
> |PCFI| 41.44 ± 7.32| 64.00 ± 8.11| 55.17 ± 12.04|55.32 ± 7.46|53.98|
> |FISF| 42.87 ± 3.02| 60.71 ± 8.75| 48.97 ± 12.22|49.47 ± 6.74| 50.51|
> |GOODIE| 36.83 ± 3.56| 63.57 ± 8.27| 38.97 ± 18.38| 45.00 ± 15.09| 46.09|
> |FSD-CAP|**43.23 ± 8.30**|**66.79 ± 6.60**|**57.24 ± 11.14**|**57.63 ± 9.00**|**56.22**|
>
>
> Fractional diffusion generalizes the normalized adjacency kernel by introducing an exponent $\gamma$ that controls the locality of propagation. Larger $\gamma$ concentrates updates on the strongest neighbors, which is suited to homophily. Smaller $\gamma$ spreads mass more uniformly across neighbors, which reduces reliance on potentially misleading local neighborhoods and is suited to heterophily. We validate this behavior on Cornell, Texas, Wisconsin, and Chameleon by tuning $\gamma$ on a validation set over [0, 2]. The best values lie in [0.2, 0.8], which confirms that milder and more distributed diffusion is preferable in low-homophily regimes.
>
> Indeed, we acknowledge that FSD-CAP is not specifically designed for heterophily graphs. In extreme heterophily scenarios, methods explicitly incorporating heterophily-aware mechanisms, such as high-order path modeling or multi-relational aggregation, may offer further gains. Exploring how to integrate such mechanisms into the diffusion-based completion pipeline constitutes an important direction for future work.
>
> **References:**
>
> [1] Zhu, Jiong, et al. "Beyond homophily in graph neural networks: Current limitations and effective designs." NIPS 2020.
>
> [2] Um, Daeho, et al. "Propagate and Inject: Revisiting Propagation-Based Feature Imputation for Graphs with Partially Observed Features." ICML 2025.
>
> [3] Yun, Sukwon, et al. "Oldie but Goodie: Re-illuminating Label Propagation on Graphs with Partially Observed Features." KDD 2025.
>
> [4] Chen, Jie, et al. "From node interaction to hop interaction: New effective and scalable graph learning paradigm." CVPR 2023.
>
> [5] Chien, Eli, et al. "Adaptive universal generalized pagerank graph neural network." ICLR 2020.

---

> ### Author Response · Authors · 2025-11-23
> **Response to Reviewer vLJV --- Part Ⅱ**
>
> **Weakness 2 & Question 2:**
>
> Thank you for your valuable suggestions. We additionally compare to feature-imputation methods that are graph-agnostic, such as setting missing features to zero (Zero), replacing them with the global mean of that feature over the graph (Global Mean), and a K-NN method that imputes missing values using the mean of similar features sampled by k-nearest neighbors (k = 5). For these baselines, as well as for our FSD-CAP, we employ GraphSage with mean aggregator [1] as the downstream GNN. The final results are reported as the mean and standard deviation over 10 runs  in the table below.
>
> **Table 1:** Accuracy(\%) for semi-supervised node classification task at 99.5\% missing rate under structural missingness. For all methods, a 2-layer GraphSage is used as downstream GNNs. The best results are highlighted in bold.
> | Method       | Cora           | CiteSeer        | PubMed          | Photo           | Computers       | Average |
> |--------------|----------------|-----------------|-----------------|-----------------|-----------------|---------|
> | Zero         | 35.75 ± 5.02   | 26.21 ± 2.14    | 43.31 ± 1.42    | 62.11 ± 4.22    | 53.63 ± 4.32    | 44.20   |
> | Global Mean  | 42.64 ± 9.79   | 31.29 ± 3.14    | 48.53 ± 4.57    | 76.51 ± 2.29    | 67.98 ± 2.75    | 53.39   |
> | K-NN         | 31.54 ± 2.34   | 24.26 ± 1.65    | 41.21 ± 1.86    | 20.20 ± 5.59    | 34.63 ± 6.33    | 30.37   |
> | FP           | 75.02 ± 1.71   | 64.42 ± 4.07    | 74.05 ± 2.40    | 85.19 ± 1.59    | 75.54 ± 1.96    | 74.84   |
> | PCFI         | 74.90 ± 1.56   | 65.97 ± 3.26    | 74.12 ± 2.81    | 86.68 ± 1.18    | 78.08 ± 1.11    | 75.95   |
> | FSD-CAP  | **77.31 ± 1.54**| **68.47 ± 2.40**| **76.19 ± 1.72**| **87.15 ± 1.67**| **78.39 ± 1.80**| **77.50**|
>
> When using GraphSAGE as the backbone, FSD-CAP consistently achieves the best performance across all datasets. Under 99.5\% structural missingness, FSD-CAP attains an average accuracy of 77.50\% across the five datasets, outperforming FP (74.84\%) and PCFI (75.95\%). Among the three generic imputation strategies, Global Mean yields better results than Zero or K-NN imputation. In addition, it can be noted that in the current experimental configuration scenario, using brute force fusion of similar neighbor node attributes to complete missing attributes is not only inferior to directly assigning missing attributes to 0, but also far inferior to using attribute mean for assignment. The results presented by these three different baseline methods fully demonstrate that when dealing with data loss issues, simple and inappropriate brute force fusion strategies based on neighbor node attributes are difficult to effectively capture the inherent characteristics and distribution patterns of data, and cannot provide reasonable and accurate supplements for missing attributes.
>
> **Reference:**
>
> [1] Hamilton, Will, Zhitao Ying, and Jure Leskovec. "Inductive representation learning on large graphs." Advances in neural information processing systems 30 (2017).

---

> > ### Comment · Reviewer_vLJV · 2025-11-25
> > **Response to the authors**
> >
> > The authors adequately addressed all my concerns.
> >
> > Specifically,
> > 1. In response to my concern related to the underlying assumptions, the authors confirmed and demonstrated that homophily is one factor affecting the node classification task and that the model identified and mitigated the effect if violated, through the evaluations on less homophilic graphs. While completely optional, it would be useful to test it on a more extreme case, like sexual networks with gender being the node attributes. This will be a useful test case to investigate the diffusion process in the heterophilic network and develop the remedy.
> >
> > 2. The authors provided comparison against standard imputation methods, demonstrating that the proposed method performed better with a substantial margin.

---

### Official Review · Reviewer_rfQB · 2025-10-31

**Soundness:** 2
**Presentation:** 3
**Contribution:** 2
**Rating:** 4
**Confidence:** 4

**Summary:**

This paper proposes FSD-CAP, a two-stage framework for imputing missing node features in graphs under high missing rates. The first stage uses fractional subgraph diffusion (FSD). The second stage applies class-aware propagation (CAP) using pseudo-labels and neighborhood entropy for feature refinement.

**Strengths:**

The fractional diffusion operator provides principled control over propagation sharpness, interpolating between uniform averaging and dominant-neighbor selection.

Progressive subgraph expansion reduces error accumulation compared to global diffusion.

Achieves competitive results across multiple datasets and missing patterns.

**Weaknesses:**

Table 11 shows counterintuitive performance gains with increasing missing rates: CiteSeer achieves 71.94% accuracy at 99.5% structural missing versus 70.00% with full features. Similar anomalies appear across multiple datasets and missing types, violating basic expectations that more missing information should degrade performance.

Critical absence of recent self-supervised graph learning methods for feature reconstruction (e.g., GRACE, GraphCL, MVGRL).

Missing comparisons with masked graph modeling approaches that follow BERT-style pre-training paradigms for graph feature completion.

No evaluation against graph contrastive learning methods (SimGRACE, GraphMAE, MaskGAE) that have shown strong performance on incomplete graphs.

Lacks comparison with recent graph pre-training methods and foundation models that can handle missing features through representation learning.

Missing recent diffusion-based graph generation methods that could serve as feature completion baselines.

Progressive subgraph expansion assumes feature similarity decreases with graph distance, but this may not hold for heterophilous graphs or when class boundaries cross multiple hops.

Convergence analysis (Theorems 2-3) relies on strong connectivity assumptions that may be violated in practice when 99.5% of features are missing.

Code availability promises may be insufficient given the complexity of the two-stage framework.

**Questions:**

The authors should address the concerns raised in the weakness section above.

---

> ### Author Response · Authors · 2025-11-23
> **Response to Reviewer rfQB --- PartⅠ(W1)**
>
> **Weaknesses:**
>
> **W1:** Thank you for highlighting this seemingly counterintuitive observation. At first glance, the phenomenon that “higher missing rates lead to better performance” appears paradoxical. However, we find that this behavior is not unique to FSD-CAP, but has also been observed in several state-of-the-art feature imputation methods, particularly on high-dimensional, sparse text graphs such as CiteSeer.
>
> For instance, when using GIN as the downstream classifier: FISF [1] achieves 69.10\% accuracy under 99.5\% structural missingness and 69.13\% under uniform missingness. PCFI [2] reaches 68.61\% under uniform missingness. In contrast, the full-feature GIN baseline on CiteSeer only attains 68.58\%, meaning that both FISF and PCFI outperform the original full-feature setting under extreme missing conditions. Similarly, when using GCN as the backbone, PCFI achieves 71.68\% accuracy at 50\% uniform missingness, surpassing the full-feature GCN baseline of 70.98\%.
>
> These results suggest that a well-designed feature imputation process not only completes missing values but also engages in attribute denoising. The original node features in CiteSeer have a high dimensionality of 3,703 but are extremely sparse, with a global sparsity rate (percentage of zero elements) as high as 99.15\%. The fact that many models achieve near-full performance with moderate missing rates indicates that the original features may contain substantial redundant information. Conversely, diffusion-based feature imputation methods like FSD-CAP, FISF, and PCFI implicitly incorporate local topology during reconstruction, thereby producing more discriminative representations than the originally observed features.
>
> Our ablation study further supports this hypothesis. On CiteSeer with a 99.5\% feature missing rate, FSD-CAP without its second stage (i.e., the Class-Aware Propagation module) achieves an accuracy of 68.65\%. Integrating this stage boosts accuracy by 3.29 points to 71.94\%, surpassing the full-feature baseline. This significant improvement indicates that the class-level prior enhances inter-class discriminability beyond basic diffusion, leading to higher-quality completed features compared to other models.
>
> **References:**
>
> [1] Um, Daeho, et al. "Propagate and Inject: Revisiting Propagation-Based Feature Imputation for Graphs with Partially Observed Features." ICML 2025.
>
> [2] Um, Daeho, et al. "Confidence-based feature imputation for graphs with partially known features." ICLR 2023.

---

> ### Author Response · Authors · 2025-11-23
> **Response to Reviewer rfQB --- Part Ⅱ (1/W2-W6)**
>
> **W2-W6:** Thank you for valuable suggestions. Methods such as GraphCL, MVGRL, GRACE, and SimGRACE are primarily designed to learn discriminative and generalizable node-level or graph-level representations via contrastive or self-supervised learning on fully observed graph data. These approaches assume that all node features are available. In particular, their loss functions operate in the representation space rather than the original feature space, and they do not involve the reconstruction of missing features. In other words, the focus is on “how to learn better representations” rather than “how to impute features under severe missingness.” Consequently, under extreme missing rates (e.g., 99.5\%), these methods are not directly applicable to the feature completion task, nor are they suitable for fair comparison with models specifically designed for high-missingness scenarios.
>
> In contrast, our work focuses on the problem of feature completion under extreme feature missingness. The core objective is to reconstruct high-quality complete feature representations from very few observed features, guided by graph structure, so as to improve the performance of downstream GNNs. This problem formulation aligns with the standard paradigm in the feature imputation subfield, as adopted by methods such as FP, PCFI, and PaGCN. Indeed, recent state-of-the-art imputation approaches, including FISF [1] and GOODIE [2], also use FP and PCFI as primary baselines. Furthermore, we include comparisons not only with these diffusion-based methods but also with distribution-matching strategies like ITR and ASD-VAE.
>
> To ensure fairness and comparability, we strictly adhere to the evaluation protocols used in prior imputation works and have now included comparative results with the latest methods, FISF and GOODIE. The results are summarized in the table below. For node classification, FSD-CAP still achieves competitive or superior performance across multiple datasets under extreme feature missing rates in both structural and uniform missing settings. At 99.5\% structural missingness, FSD-CAP achieves an average accuracy of 80.06\% across the five datasets, which is higher than FISF's 78.27\% and GOODIE’s 74.67\%. Under uniform missingness, FSD-CAP achieves an average accuracy of 81.01\%, outperforming FISF (79.12\%) and GOODIE (77.06\%).
>
> **Table1:** Accuracy(\%) for semi-supervised node classification task at 99.5\% missing rate. The best results are highlighted in bold.
>
> Structural Missing:
> | Method| Cora| CiteSeer| PubMed| Photo| Computers| Average |
> |------------|----------------|-----------------|-----------------|-----------------|-----------------|---------|
> | FISF| 79.01 ± 1.79   | 69.71 ± 2.49    | 76.21 ± 1.07    | 87.86 ± 1.24    | 78.55 ± 1.47    | 78.27   |
> | GOODIE| 74.40 ± 2.31   | 61.58 ± 4.38    | 72.81 ± 2.56    | 86.65 ± 1.16    | 77.92 ± 2.55    | 74.67   |
> | FSD-CAP| **80.56 ± 1.83**| **71.94 ± 1.32**| **76.98 ± 1.41**| **89.18 ± 0.97**| **81.64 ± 1.21**| **80.06**|
>
> Uniform Missing:
> | Method| Cora| CiteSeer| PubMed| Photo| Computers| Average |
> |------------|----------------|-----------------|-----------------|-----------------|-----------------|---------|
> | FISF| 79.27 ± 1.49   | 69.16 ± 2.60    | **77.47 ± 1.00**| 88.12 ± 1.33    | 81.58 ± 0.27    | 79.12   |
> | GOODIE| 78.02 ± 1.28   | 65.71 ± 1.47    | 74.00 ± 1.73    | 87.60 ± 1.70    | 79.99 ± 1.36    | 77.06   |
> | FSD-CAP| **81.49 ± 1.95**| **73.15 ± 0.98**| 77.46 ± 1.15    | **89.40 ± 0.94**| **83.57 ± 0.95**| **81.01**|

---

> ### Author Response · Authors · 2025-11-23
> **Response to Reviewer rfQB --- Part Ⅱ (2/W2-W6)**
>
> **W2-W6**:
>
> For link prediction task, we observed a notable discrepancy between the results obtained by running FISF with the hyperparameters reported in paper and the performance figures reported therein. To ensure a fair comparison, we include both our reproduced results and the performance from the FISF paper (denoted as FISF-paper) in our evaluation. Even when comparing against the best-reported results in FISF, FSD-CAP consistently achieves superior performance. Specifically, under structural missingness, FSD-CAP achieves an average AUC score of 91.65\%, outperforming FISF-paper (89.03\%) by 2.62\%. In the uniform missing setting, FSD-CAP achieves 92.41\% compared to 89.60\% for FISF-paper. These results demonstrate the effectiveness and robustness of FSD-CAP in extreme missing scenarios.
>
> **Table1:** Performance on link prediction tasks at 99.5\% missing rate, measured by ROC AUC score (\%). The best results are highlighted in bold.
>
> Structural Missing:
> | Method| Cora| CiteSeer| PubMed| Photo| Computers| Average |
> |--------------|----------------|-----------------|-----------------|------------------|------------------|---------|
> | FISF| 48.56 ± 0.81   | 54.31 ± 0.97    | 56.70 ± 0.46    | 55.19 ± 10.37    | 57.85 ± 8.86     | 54.52   |
> | FISF-paper| 87.26 ± 1.44   | 84.12 ± 1.17    | 83.19 ± 0.78    | 95.86 ± 0.21     | 94.70 ± 0.30     | 89.03   |
> | GOODIE| 63.40 ± 2.05   | 63.42 ± 1.26    | 72.03 ± 2.13    | 93.01 ± 0.87     | 71.87 ± 1.63     | 72.75   |
> | FSD-CAP| **87.97 ± 1.24**| **87.48 ± 0.96**| **88.39 ± 0.57**| **97.55 ± 0.16** | **96.85 ± 0.13** | **91.65**|
>
> Uniform Missing:
> | Method| Cora| CiteSeer| PubMed| Photo| Computers| Average |
> |--------------|----------------|-----------------|-----------------|------------------|------------------|---------|
> | FISF| 48.11 ± 1.16   | 52.65 ± 0.51| 57.28 ± 0.79| 58.41 ± 0.23     | 58.52 ± 0.07     | 54.99   |
> | FISF-paper| 87.44 ± 0.80   | 83.45 ± 2.53| 85.33 ± 0.47| 96.64 ± 0.18     | 95.13 ± 0.35     | 89.60   |
> | GOODIE| 64.70 ± 0.99   | 66.32 ± 0.74    | 71.87 ± 1.63    | 94.80 ± 0.23     | 92.55 ± 0.49     | 78.05   |
> | FSD-CAP| **90.01 ± 1.10**| **87.70 ± 1.09**| **88.50 ± 0.45**| **98.16 ± 0.05** | **97.69 ± 0.07** | **92.41**|
>
> We fully acknowledge the strong potential of graph contrastive learning, masked modeling, and other self-supervised paradigms in representation learning. A promising direction would be to integrate such representation learning ideas into the feature completion process, which is a key area we plan to explore in our follow-up research.
>
> References:
>
> [1] Um, Daeho, et al. "Propagate and Inject: Revisiting Propagation-Based Feature Imputation for Graphs with Partially Observed Features." ICML 2025.
>
> [2] Yun, Sukwon, et al. "Oldie but Goodie: Re-illuminating Label Propagation on Graphs with Partially Observed Features." KDD 2025.

---

> ### Author Response · Authors · 2025-11-23
> **Response to Reviewer rfQB --- Part Ⅲ (W7-W8)**
>
> **W7:** Progressive subgraph expansion assumes feature similarity decreases with graph distance, but this may not hold for heterophilous graphs or when class boundaries cross multiple hops.
>
> **Response:**
> While most diffusion-based methods implicitly rely on the homophily assumption, the proposed FSD-CAP mitigates the strong dependence on local features by integrating a fractional diffusion process with a class-aware feature refinement mechanism. To evaluate its performance on heterophily-dominant graphs, we conduct experiments on four standard heterophilic benchmarks: Cornell, Texas, Wisconsin (WebKB), and Chameleon (Wikipedia). These graphs exhibit low edge homophily ratios of 0.30, 0.11, 0.21, and 0.23, respectively, where edge homophily is defined as the fraction of edges that connect nodes of the same class.
>
> We test node classification at a 99.5\% missing rate under both structural and uniform settings. We compare with diffusion based completion methods FP and PCFI, and with two state-of-the-art GNNs designed for heterophily, namely, Hop-GNN and GPR-GNN, under the same missingness. Hop-GNN uses preprocessed multi hop features inside each node. GPR-GNN learns adaptive PageRank weights to extract features and topology without assuming homophily.
>
> The table in Appendix A.2.8 reports accuracies on the four heterophily graphs. When features are extremely sparse, the heterophily GNNs degrade, while completion first helps by reconstructing informative representations. FSD-CAP is superior across all datasets. Under structural missingness, Hop-GNN averages 44.97\% and FSD-CAP reaches 56.22\%. Compared with PCFI, FSD-CAP improves accuracy by 1.79\% on Chameleon, 2.79\% on Texas, 2.07\% on Cornell, and 2.31\% on Wisconsin. Under uniform missingness, the advantage grows. FSD-CAP averages 58.85\%. It exceeds PCFI by 7.30\% on Chameleon, 8.57\% on Texas, 1.69\% on Cornell, and 1.58\% on Wisconsin.
>
> Fractional diffusion generalizes the normalized adjacency kernel by introducing an exponent $\gamma$ that controls the locality of propagation. Larger $\gamma$ concentrates updates on the strongest neighbors, which is suited to homophily. Smaller $\gamma$ spreads mass more uniformly across neighbors, which reduces reliance on potentially misleading local neighborhoods and is suited to heterophily. We validate this behavior on Cornell, Texas, Wisconsin, and Chameleon by tuning $\gamma$ on a validation set over [0, 2]. The best values lie in [0.2, 0.8], which confirms that milder and more distributed diffusion is preferable on heterophilous graphs.
>
> **W8:** Convergence analysis (Theorems 2-3) relies on strong connectivity assumptions that may be violated in practice when 99.5\% of features are missing.
>
> **Response:**
> We sincerely apologize for any confusion caused. In this work, both feature missingness patterns, structural missing and uniform missing, affect only the observability of node features, not the underlying graph topology. Specifically, under structural missingness, all features of certain nodes are masked as unknown, but their adjacency relationships (i.e., connections to neighbors) remain intact.
> Under uniform missingness, a random subset of feature dimensions is masked for nodes, while the remaining features and all graph edges are fully preserved. Consequently, the adjacency matrix $A$ and the overall graph connectivity remain unchanged regardless of the feature missing rate, even at extreme levels such as 99.5\%. Our model consistently performs subgraph diffusion based on the original graph structure.
>
> In Theorem 2, $G^{(m)}$ denotes the m-hop subgraph centered on observed nodes, constructed solely based on the original adjacency relations, independent of feature availability. Even under extreme missing rates (e.g., 99.5\%), as long as the graph remains connected, the subgraph expansion mechanism described in Theorem 3 can still progressively cover the entire graph and asymptotically approximate the global diffusion solution.
>
> Therefore, the connectivity assumption required for our convergence analysis still holds in our setting, since feature missingness does not compromise the topological integrity of the graph.

---

> ### Author Response · Authors · 2025-11-23
> **Response to Reviewer rfQB --- Part Ⅳ (W9)**
>
> **W9:** Code availability promises may be insufficient given the complexity of the two-stage framework.
>
>
> **Response:**
> Thank you for this insightful question. FSD-CAP has two stages. The first (progressive subgraph diffusion) dominates runtime of feature imputation. It performs iterative diffusion over a small number of expanded subgraphs. The second (class-aware refinement) is a single propagation on a compact class-level graph and adds negligible overhead.
>
> In the progressive stage, we use efficient sparse matrix multiplications to update node features at each diffusion step and each subgraph layer. In practice, 100 iterations are sufficient for convergence across all datasets, which balances accuracy and efficiency. Only 4 to 7 expansion rounds are needed to cover the graph effectively. For example, On Cora, one fractional-diffusion pass on a subgraph (100 iterations) takes approximately 60 ms, with six rounds take 380 ms in total. The subsequent class-aware refinement adds only about 6 ms, resulting in an imputation time of less than 0.4 seconds.
>
> We report wall-clock time (in seconds), including both the time for feature imputation and training of downstream GCN for FSD-CAP and baseline models in the table below. As shown in the table, learning-based methods incur significantly higher computational costs compared to diffusion-based imputation approaches. For FSD-CAP, the vast majority of the time is spent on training the downstream GCN. The actual feature completion stage, in contrast, is highly efficient, requiring only 0.38s on Cora, 0.76s on CiteSeer, 0.75s on PubMed, 0.57s on Photo, and 1.09s on Computers. Among diffusion-based models, FSD-CAP requires more imputation time than FP and PCFI, yet the cost remains moderate. The extra time comes from the multi step subgraph expansion, which improves reconstruction under extreme missingness. With only 0.5\% of features available, FSD-CAP improves over PCFI by 1.80\% to 5.88\% on Cora, CiteSeer, PubMed, Photo, and Computers, with an average gain of 3.67\%. This yields a favorable trade-off between accuracy and efficiency and is practical for real-world settings with severe feature scarcity.
>
> **Table1:** Training time of methods under structural 99.5% missing rate.
> | Method   | Cora     | CiteSeer | PubMed   | Photo    | Computers |
> | -------- | -------- | -------- | -------- | -------- | --------- |
> | PaGCN    | 3.442s   | 6.035s   | 11.297s  | 12.643s  | 26.635s   |
> | GRAFENNE | 78.285s  | 30.133s  | 225.987s | 160.739s | 223.767s  |
> | ASD-VAE  | 328.785s | 132.977s | OOM      | 468.35s  | OOM       |
> | FP       | 2.251s   | 2.282s   | 2.911s   | 3.267s   | 7.545s    |
> | PCFI     | 2.959s   | 2.797s   | 3.763s   | 3.769s   | 8.327s    |
> | FSD-CAP  | 3.276s   | 3.103s   | 4.022s   | 4.329s   | 8.992s    |

---

### Official Review · Reviewer_66Pn · 2025-10-31

**Soundness:** 2
**Presentation:** 3
**Contribution:** 2
**Rating:** 2
**Confidence:** 4

**Summary:**

This paper proposes FSD-CAP (Fractional Subgraph Diffusion with Class-Aware Propagation), a diffusion-based framework for imputing missing node features in graphs with partially observed features. The first stage, fractional subgraph diffusion, introduces a tunable fractional operator that adjusts propagation sharpness according to local graph structure, and progressively expands diffusion from observed to unobserved nodes to limit error accumulation. The second stage, class-aware propagation, refines the imputed features using pseudo-labels and neighborhood-entropy weighting to promote intra-class consistency and inter-class separation. Experiments demonstrate the effectiveness of FSD-CAP on semi-supervised node classification and link prediction at a high missing rate.

**Strengths:**

1. The authors present a new diffusion mechanism for imputation on graph-structured data.
2. The paper is clearly written and organized.

**Weaknesses:**

1. State-of-the-art methods [1, 2] are missing from the comparison. It appears that the state-of-the-art methods [1,2] achieve better performance than the proposed method, raising questions about the claimed performance advantages. Futhermore, [2] adopts a similar idea of leveraging label information during propagation.
2. Since the proposed method leverages label information, its performance is likely to depend on label availability; however, no experiments are provided to analyze sensitivity to different label rates.
3. In link prediction, it is generally assumed that only graph connectivity is available, but the paper additionally utilizes node labels.
4. The motivation or empirical observation behind the design of the Fractional Diffusion Operator is insufficiently explained.
5. It would be preferable to represent matrices and vectors in italicized bold.

[1] Um, Daeho, et al. "Propagate and Inject: Revisiting Propagation-Based Feature Imputation for Graphs with Partially Observed Features." ICML 2025.\
[2] Yun, Sukwon, et al. "Oldie but Goodie: Re-illuminating Label Propagation on Graphs with Partially Observed Features." KDD 2025.

**Questions:**

Q1. Could the authors provide performance comparisons with state-of-the-art methods [1, 2] for both node classification and link prediction tasks?

Q2. Since the proposed method leverages label information, could the authors show how performance changes when the label rate is lower than the current experimental setting?

Q3. The Fractional Diffusion Operator appears to down-weight the participation of nodes with higher degrees during diffusion. Could the authors clarify the motivation or empirical rationale (not resulting performance) behind this design choice?

---

> ### Author Response · Authors · 2025-11-23
> **Response to Reviewer 66Pn --- PartⅠ**
>
> **Weaknesses:**
>
> **W1:** Please refer to our answer to Question 1.
>
> **W2:** Please refer to our answer to Question 2.
>
> **W3:** In standard link prediction, models typically leverage graph structure to construct positive and negative edges combined with node features for binary classification. However, recent approaches, such as CGLE [1], Pro-SEAL [2] and the SOTA completion method GOODIE [3] mentioned in the Weakness review, have also integrated class-level semantic information by leveraging labels to provide node-specific attributes, thereby enhancing representation learning and link prediction performance.
>
> To validate the effectiveness of our method in label-free settings, we remove the second stage of FSD-CAP, the Class-Aware Propagation module that constructs a class-level graph using label information, and retain only the first stage --- Fractional Subgraph Diffusion (i.e., FSD-only). On five benchmarks under 99.5\% feature missingness, the simplified variant still achieves strong performance on link prediction, with average AUC scores of 90.43\%, slightly lower than the full FSD-CAP (91.65\%). This demonstrates that our framework can effectively reconstruct high-quality node features without utilizing label information, enabling robust downstream link prediction.
>
> Furthermore, to ensure a fair comparison when label information is available, we compare against GOODIE, a state-of-the-art method that similarly leverages labels to assist feature completion. We adopt the settings from their released code, and the result shows that FSD-CAP achieves an average AUC of 91.65\% across five datasets, compared to 72.75\% for GOODIE. This improvement highlights that the proposed class-level refinement more effectively integrates class-level priors, yielding higher-quality node representations for link prediction when labels are accessible.
>
> **Table1:** Performance on link prediction tasks at 99.5\% missing rate under structural missingness, measured by ROC AUC score (\%). The best results are highlighted in bold.
> | Method| Cora| CiteSeer| PubMed| Photo| Computers| Average|
> |------------|----------------|-----------------|-----------------|-----------------|-----------------|---------|
> | FSD-only| 87.20 ± 1.36   | 86.85 ± 1.25    | 87.42 ± 0.39    | 94.61 ± 3.47    | 96.06 ± 0.20    | 90.43   |
> | FSD-CAP| **87.97 ± 1.24**| **87.48 ± 0.96**| **88.39 ± 0.57**| **97.55 ± 0.16**| **96.85 ± 0.13**| **91.65**|
> | GOODIE| 63.40 ± 2.05   | 63.42 ± 1.26    | 72.03 ± 2.13    | 93.01 ± 0.87    | 71.87 ± 1.63    | 72.75   |
>
> **References:**
>
> [1] Mazumder, Ankit, and Srikanta Bedathur. "CGLE: Class-label Graph Link Estimator for Link Prediction." arXiv preprint arXiv:2511.06982 (2025). ICDM 2025.
>
> [2] Xiong, Tianyu, Chenyang Qiu, and Peng Zhang. "How ground-truth label helps link prediction in heterogeneous graphs." ADMIT 2023.
>
> [3] Yun, Sukwon, et al. "Oldie but Goodie: Re-illuminating Label Propagation on Graphs with Partially Observed Features." KDD 2025.
>
> **W4:** Please refer to our answer to Question 3.
>
> **W5:** Thank you for your valuable suggestion. The notation has been revised accordingly to ensure consistency with standard typographical conventions.

---

> ### Author Response · Authors · 2025-11-23
> **Response to Reviewer 66Pn --- Part Ⅱ (1/Q1)**
>
> **Questions:**
>
> **Q1:** Thank you for your valuable feedback. We have included comparisons with FISF [1] and GOODIE [2] for both node classification and link prediction tasks in our revised experiments. For baselines, we adopt the settings from the authors’ released code or papers. The results are summarized in tables below.
>
> For node classification, FSD-CAP still achieves competitive or superior performance across multiple datasets under extreme feature missing rates in both structural and uniform missing settings. At 99.5\% structural missingness, FSD-CAP achieves an average accuracy of 80.06\% across the five datasets, which is higher than FISF's 78.27\% and GOODIE’s 74.67\%. Under uniform missingness, FSD-CAP achieves an average accuracy of 81.01\%, outperforming FISF (79.12\%) and GOODIE (77.06\%).
>
> **Table1:** Accuracy(\%) for semi-supervised node classification task at 99.5\% missing rate. The best results are highlighted in bold.
>
> Structural Missing:
> | Method| Cora| CiteSeer| PubMed| Photo| Computers| Average |
> |------------|----------------|-----------------|-----------------|-----------------|-----------------|---------|
> | FISF| 79.01 ± 1.79   | 69.71 ± 2.49    | 76.21 ± 1.07    | 87.86 ± 1.24    | 78.55 ± 1.47    | 78.27   |
> | GOODIE| 74.40 ± 2.31   | 61.58 ± 4.38    | 72.81 ± 2.56    | 86.65 ± 1.16    | 77.92 ± 2.55    | 74.67   |
> | FSD-CAP| **80.56 ± 1.83**| **71.94 ± 1.32**| **76.98 ± 1.41**| **89.18 ± 0.97**| **81.64 ± 1.21**| **80.06**|
>
> Uniform Missing:
> | Method| Cora| CiteSeer| PubMed| Photo| Computers| Average |
> |------------|----------------|-----------------|-----------------|-----------------|-----------------|---------|
> | FISF| 79.27 ± 1.49   | 69.16 ± 2.60    | **77.47 ± 1.00**| 88.12 ± 1.33    | 81.58 ± 0.27    | 79.12   |
> | GOODIE| 78.02 ± 1.28   | 65.71 ± 1.47    | 74.00 ± 1.73    | 87.60 ± 1.70    | 79.99 ± 1.36    | 77.06   |
> | FSD-CAP| **81.49 ± 1.95**| **73.15 ± 0.98**| 77.46 ± 1.15    | **89.40 ± 0.94**| **83.57 ± 0.95**| **81.01**|
>
> For the link prediction task, we observed a notable discrepancy between the results obtained by running FISF with the hyperparameters reported in paper and the performance figures reported therein. To ensure a fair comparison, we include both our reproduced results and the performance from the FISF paper (denoted as FISF-paper) in our evaluation. Even when comparing against the best-reported results from the FISF paper, FSD-CAP consistently achieves superior performance. Specifically, under structural missingness, FSD-CAP achieves an average AUC score of 91.65\%, outperforming FISF-paper (89.03\%) by 2.62\%. In the uniform missing setting, FSD-CAP achieves 92.41\% compared to 89.60\% for FISF-paper. These results demonstrate the effectiveness and robustness of our method in extreme missing scenarios.
>
> **Table2:** Performance on link prediction tasks at 99.5\% missing rate, measured by ROC AUC score (\%). The best results are highlighted in bold.
>
> Structural Missing:
> | Method| Cora| CiteSeer| PubMed| Photo| Computers| Average |
> |--------------|----------------|-----------------|-----------------|------------------|------------------|---------|
> | FISF| 48.56 ± 0.81   | 54.31 ± 0.97    | 56.70 ± 0.46    | 55.19 ± 10.37    | 57.85 ± 8.86     | 54.52   |
> | FISF-paper| 87.26 ± 1.44   | 84.12 ± 1.17    | 83.19 ± 0.78    | 95.86 ± 0.21     | 94.70 ± 0.30     | 89.03   |
> | GOODIE| 63.40 ± 2.05   | 63.42 ± 1.26    | 72.03 ± 2.13    | 93.01 ± 0.87     | 71.87 ± 1.63     | 72.75   |
> | FSD-CAP| **87.97 ± 1.24**| **87.48 ± 0.96**| **88.39 ± 0.57**| **97.55 ± 0.16** | **96.85 ± 0.13** | **91.65**|
>
> Uniform Missing:
> | Method| Cora| CiteSeer| PubMed| Photo| Computers| Average |
> |--------------|----------------|-----------------|-----------------|------------------|------------------|---------|
> | FISF| 48.11 ± 1.16   | 52.65 ± 0.51| 57.28 ± 0.79| 58.41 ± 0.23     | 58.52 ± 0.07     | 54.99   |
> | FISF-paper| 87.44 ± 0.80   | 83.45 ± 2.53| 85.33 ± 0.47| 96.64 ± 0.18     | 95.13 ± 0.35     | 89.60   |
> | GOODIE| 64.70 ± 0.99   | 66.32 ± 0.74    | 71.87 ± 1.63    | 94.80 ± 0.23     | 92.55 ± 0.49     | 78.05   |
> | FSD-CAP| **90.01 ± 1.10**| **87.70 ± 1.09**| **88.50 ± 0.45**| **98.16 ± 0.05** | **97.69 ± 0.07** | **92.41**|
>
> **References:**
>
> [1] Um, Daeho, et al. "Propagate and Inject: Revisiting Propagation-Based Feature Imputation for Graphs with Partially Observed Features." ICML 2025.
>
> [2] Yun, Sukwon, et al. "Oldie but Goodie: Re-illuminating Label Propagation on Graphs with Partially Observed Features." KDD 2025.

---

> ### Author Response · Authors · 2025-11-23
> **Response to Reviewer 66Pn --- Part Ⅱ (2/Q1)**
>
> **Q1:**
>
> It is worth noting that GOODIE combines label propagation (LP) and feature propagation (FP) through an attention mechanism, and further leverages pseudo-labels generated from the LP branch for contrastive learning. This design implicitly assumes that connected nodes tend to share the same label, i.e., strong label homophily.
>
> We test node classification at a 99.5% missing rate under both structural and uniform settings on four standard heterophilic benchmarks: Cornell, Texas, Wisconsin (WebKB), and Chameleon (Wikipedia). Their edge homophily ratios are 0.30, 0.11, 0.21, and 0.23, defined as the fraction of edges that connect nodes of the same class.
>
> As shown in table, on heterophily graphs, due to unreliable pseudo-labels from label propagation, GOODIE’s performance drops notably. In contrast, FSD-CAP achieves consistently better results among recent state-of-the-art methods on these datasets, indicating robust performance across diverse graph structures. In structural missing setting, GOODIE averages 46.09\%, which is 10.13\% below FSD-CAP result of 56.22\%. Under uniform missingness, the performance gap increases to 12.87\%. These findings are consistent with our design. Progressive subgraph diffusion limits early cross-class leakage, and fractional diffusion allows milder mixing on weak homophily. The class-aware refinement then pulls uncertain nodes toward class prototypes and downweights boundary nodes by entropy.
>
> **Table1:** Accuracy(\%) on heterophily graphs for semi-supervised node classification task at 99.5\% missing rate. The best results are highlighted in bold.
>
> Structural Missing:
> |Method|Chameleon|Texas|Cornell|Wisconsin|Average|
> |------------|----------------|-----------------|-----------------|-----------------|---------|
> |Hop-GNN| 26.65 ± 9.66|57.11 ± 6.56|47.11 ± 6.56|49.02 ± 5.95|44.97|
> |GPR-GNN| 21.90 ± 1.75|58.42 ± 6.09|48.42 ± 6.09|46.08 ± 5.49|43.70|
> |FP|39.40 ± 2.97| 61.07 ± 8.81| 55.36 ± 12.12| 55.53 ± 7.20|52.84|
> |PCFI| 41.44 ± 7.32| 64.00 ± 8.11| 55.17 ± 12.04|55.32 ± 7.46|53.98|
> |FISF| 42.87 ± 3.02| 60.71 ± 8.75| 48.97 ± 12.22|49.47 ± 6.74| 50.51|
> |GOODIE| 36.83 ± 3.56| 63.57 ± 8.27| 38.97 ± 18.38| 45.00 ± 15.09| 46.09|
> |FSD-CAP|**43.23 ± 8.30**|**66.79 ± 6.60**|**57.24 ± 11.14**|**57.63 ± 9.00**|**56.22**|
>
> Uniform Missing:
> | Method| Chameleon| Texas| Cornell| Wisconsin| Average|
> |------------|----------------|-----------------|-----------------|-----------------|---------|
> | Hop-GNN| 25.19 ± 6.75| 57.11 ± 6.56| 47.11 ± 6.56| 49.02 ± 5.95| 44.61|
> | GPR-GNN| 23.00 ± 1.68| 58.42 ± 6.09| 48.42 ± 6.09| 46.08 ± 5.49| 43.98|
> | FP| 40.36 ± 4.47| 61.43 ± 8.57| 55.86 ± 12.12| 58.16 ± 4.92| 53.95|
> | PCFI| 40.54 ± 4.08| 62.50 ± 9.75| 55.86 ± 12.12| 57.37 ± 6.21| 54.07|
> | FISF| 40.84 ± 1.83| 60.00 ± 9.42| 48.97 ± 7.03| 47.89 ± 3.07| 49.43|
> | GOODIE| 36.29 ± 3.42| 63.21 ± 8.15| 38.62 ± 17.90| 45.79 ± 15.93| 45.98|
> | FSD-CAP| **47.84 ± 5.21**| **71.07 ± 8.06**| **57.55 ± 11.66**| **58.95 ± 5.79**| **58.85**|

---

> ### Author Response · Authors · 2025-11-23
> **Response to Reviewer 66Pn --- Part Ⅲ（Q2）**
>
> **Q2:** Thank you for this insightful suggestion. In our original experiments, we followed the semi-supervised setting adopted in prior work (FP [1]), which assigns 20 labeled nodes per class. To evaluate the impact of label availability, we have conducted additional experiments across multiple datasets under varying label rates. Specifically, we test scenarios where each class is provided with $5, 10, 20, 30$, and $40$ labeled nodes, under three feature missing rates: 70%, 90%, and 99.5% missingness. Results on Cora and PubMed are summarized in the table below, reporting semi-supervised node classification accuracy under the structural missing setting.
>
> **Table1:** Accuracy(\%) of FSD-CAP on Photo for node classification task under varying feature missing rates and label rates.
>
> Structural Missing:
> | | 5| 10| 20| 30| 40|
> |--------------|----------------|----------------|----------------|----------------|----------------|
> | 70% missing  | 76.41 ± 3.21   | 78.42 ± 1.29   | 80.49 ± 1.86   | 81.33 ± 1.97   | 81.39 ± 1.86   |
> | 90% missing  | 75.59 ± 2.98   | 78.25 ± 1.73   | 80.27 ± 1.16   | 81.43 ± 1.54   | 81.27 ± 1.16   |
> | 99.5% missing| 74.49 ± 2.94   | 78.07 ± 1.87   | 79.55 ± 1.57   | 79.89 ± 2.01   | 79.95 ± 1.65   |
>
> Uniform Missing:
> | | 5| 10| 20| 30| 40|
> |--------------|----------------|----------------|----------------|----------------|----------------|
> | 70% missing  | 86.84 ± 1.98   | 88.18 ± 1.71   | 91.35 ± 0.56   | 91.57 ± 0.35   | 91.43 ± 0.48   |
> | 90% missing  | 86.39 ± 1.98   | 87.01 ± 2.08   | 90.60 ± 0.54   | 91.17 ± 0.44   | 90.80 ± 0.52   |
> | 99.5% missing| 84.52 ± 2.32   | 84.93 ± 2.19   | 88.98 ± 1.01   | 89.42 ± 0.92   | 88.88 ± 0.97   |
>
> The results show that the performance of FSD-CAP consistently improves as the number of labeled nodes per class increases from 5 to 30, and the gains begin to saturate when reaching 30–40 labels per class, indicating that the model has effectively leveraged the available supervision and approached convergence under the current configuration.
>
> We further evaluate FSD-CAP in an extreme low-label setting (five labels per class), combined with 99.5\% feature missingness. We compare FSD-CAP against several recent state-of-the-art baselines in this challenging setting. As shown in the Table 2, while all methods suffer performance degradation due to the scarcity of supervision signals, FSD-CAP still significantly outperforms all competitors. This demonstrates the effectiveness of the class-level prior construction, which enables more robust propagation even when labeled data are extremely limited.
>
> **Table2:** Classification accuracy (\%) under 99.5\% structural missingness and 5 labeled nodes per class. The best results are highlighted in bold.
> | Method     | Cora           | CiteSeer        | PubMed          | Photo           | Computers       | Average |
> |------------|----------------|-----------------|-----------------|-----------------|-----------------|---------|
> | FP         | 74.45 ± 3.16   | 67.05 ± 2.64    | 64.71 ± 9.10    | 82.01 ± 3.06    | 71.07 ± 4.56    | 71.86   |
> | PCFI       | 67.76 ± 3.46   | 58.68 ± 4.55    | 66.91 ± 6.46    | 81.26 ± 2.02    | 73.30 ± 2.09    | 69.58   |
> | FISF       | 73.69 ± 2.93   | 63.10 ± 2.94    | 73.27 ± 1.93    | 82.12 ± 2.68    | 73.05 ± 2.21    | 73.05   |
> | GOODIE     | 62.37 ± 4.51   | 52.00 ± 4.74    | 62.35 ± 7.20    | 80.59 ± 3.92    | 72.97 ± 2.37    | 66.06   |
> | FSD-CAP| **74.49 ± 2.94**| **68.26 ± 2.59**| **68.79 ± 3.13**| **84.52 ± 2.32**| **75.71 ± 1.86**| **74.35**|
>
> **Reference:**
>
> [1] Rossi, Emanuele, et al. "On the unreasonable effectiveness of feature propagation in learning on graphs with missing node features." Learning on graphs conference. PMLR, 2022.

---

> ### Author Response · Authors · 2025-11-23
> **Response to Reviewer 66Pn --- Part Ⅳ（Q3）**
>
> **Q3:** The design of the fractional diffusion operator is inspired by epidemic models such as the Susceptible–Infected–Recovered (SIR) model [1]. In such models, disease spread does not occur through uniform mixing of the entire population. Instead, it originates from a few initial infection sources, forming expanding "infected region". The rate of new infections is primarily governed by the size of the boundary between infected and susceptible regions. Empirical and theoretical studies show that this boundary scales as a fractional power of the infected population size. Furthermore, recent work shows that contagion can also be amplified through higher-order group interactions, where simultaneous exposure to multiple infected neighbors leads to super-linear infection dynamics [2].
>
> In a similar spirit to the SIR model, observed (feature-complete) nodes serve as “sources” of information, analogous to initially infected individuals. The goal is to propagate their information through the graph structure to reconstruct missing features. Just as in epidemiology, the efficiency of this propagation depends on the “boundary characteristics” between known and unknown regions of the graph.
>
> Conventional graph diffusion operators typically perform degree-normalized averaging, which tends to homogenize node representations and leads to over-smoothing. To address this, we introduce a fractional diffusion operator that modulates the sharpness of propagation.
> The fractional diffusion operator $A^{\gamma}$, defined by element-wise exponentiation and row-wise normalization:
>
> $A_{ij}^\gamma=\frac{(1/\sqrt{d_id_j})^\gamma}{\sum_{k\in\mathcal{N}(i)}(1/\sqrt{d_id_k})^\gamma}=\frac{(d_j)^{-\gamma/2}}{\sum_{k\in\mathcal{N}(i)}(d_k)^{-\gamma/2}}$
>
> This operator applies a nonlinear transformation to the normalized adjacency matrix, thereby amplifying structural disparities in local neighborhoods while preserving the row-stochastic property to ensure numerical stability during diffusion.
>
> Indeed, the fractional operator does not simply suppress high-degree nodes. Rather, it redefines the logic of weight allocation. Under this scheme, when a high-degree node acts as an information sender, its influence on neighbors is further suppressed due to the nonlinear enhancement of degree-normalized. When it acts as an information receiver, it becomes more sensitive to signals from low-degree neighbors, helping preserve local variation and mitigate over-smoothing.
>
> **Reference:**
>
> [1] Taghvaei, Amirhossein, et al. "Fractional SIR epidemiological models." Scientific reports 10.1 (2020): 20882.
>
> [2] Iacopini, Iacopo, et al. "Simplicial models of social contagion." Nature communications 10.1 (2019): 2485.

---

### Official Review · Reviewer_Wp9X · 2025-11-01

**Soundness:** 3
**Presentation:** 3
**Contribution:** 3
**Rating:** 6
**Confidence:** 4

**Summary:**

The authors present FSD-CAP, a novel method for missing feature imputation within graph representation learning by applying a 1) Factional diffusion operator, 2) Progressive subgraph diffusion, and 3) Class-level feature refinement. The authors introduce the mechanisms through a series of propositions and theorems that details the nuance between needing to extrapolate from coarser to finer features within subgraph diffusion. Notably, FSD-CAP outperforms the current SOTA method PCF1 by a fairly-significant margin and shows improvement over standard baselines on the CiteSeer dataset.

**Strengths:**

1) The authors provide concise yet clear details about the uniqueness of the operators and diffusion refinement stages for FSD-CAP. Further explanations detail clarifying insights about the practical considerations from moving between localized and distant updates.
2) The writing of the paper flows well, transitions between sections make sense and are not abrupt. This eases understanding of more technical concepts.
3) FSD-CAP significantly outperforms current SOTA methods like PCF1, indicating a potentially broad-level of promise for subgraph diffusion models in other applications of graph representation learning.

**Weaknesses:**

1) Although this is common within existing graph-diffusion feature imputation literature, FSD-CAP only tests on Cora, CiteSeer, PubMed, and Amazon-Photo, Amazon-Computers. Further tests on smaller synthetic datasets with defined local and global structures could indicate the power of Theorem 2 in potentially more-difficult scenarios
2) There are no algorithms within the paper to describe the process of FSD-CAP. Although this is partially-abated by the inclusion of code, the pseudo-code seems important for clarity when detailing how the equations within the paper function in-practice.

**Questions:**

* In regards to Remark 1, is a super-diffusion regime something that other diffusion models remain in since they do not have a class-refinement mechanism?
* Lines 184-187: Given that much of this is step-by-step instructions of FSD-CAP function, this seems better formatted in an algorithm?
* Lines 214-215: equation 3, equation 4 ==> uppercase?
* Theorem 2: The idea that you need Equation 3 for updates and Equation 4 for maintaining stability makes sense. However, I am confused by the notation in Theorem 2. Does $x^{(x)}(t)$ apply both Equation 3 and Equation 4 per-layer in sequence (Equation 3 - top to Equation 4 - bottom), as expected with piece-wise functions (indicated by the brackets)? I can certainly see that it would be a 'layer-wise' function, but it still seems confusing that $x^{(x)}(t)$ is the same regardless of the right-hand side of the function. This could be a mistake on my part as a reviewer, but I want to clarify with the authors, is this intended? If so, wouldn't an algorithmic description provide more clarity on the order of operations?

---

> ### Author Response · Authors · 2025-11-23
> **Response to Reviewer Wp9X --- Part I**
>
> **Weaknesses:**
>
> **W1:** Thank you for the valuable feedback. To further validate the effectiveness of Theorem 2 under more complex and challenging local and global structural settings, we constructed a synthetic dataset derived from the PubMed citation network as suggested. It preserves the semantic integrity of node features while include diverse structural patterns.
> Our construction starts with the largest connected component (LCC) extracted from the original PubMed graph to ensure topological integrity. The graph structure comprises three distinct subgraphs, each containing 200 nodes and generated using unique strategies.
> - **Original-topology subgraph:** A connected 200-node subgraph sampled directly from the original PubMed LCC. This component preserves the intrinsic topological and relational characteristics of the real-world citation network.
> - **Attribute-similarity subgraph:** A subgraph where edges are formed based on node feature cosine similarity, with a connection established only if the similarity exceeds 0.5. If the resulting graph is disconnected, a minimum spanning tree (MST) is added to ensure connectivity. This emphasizes attribute-affinity over original topology.
> - **Random-structure subgraph:** A subgraph generated using the Erdős–Rényi random graph model with a fixed edge probability p=0.2. MST is also incorporated if needed to guarantee connectivity, thereby introducing structural heterophily and breaking inherent homophily tendencies.
>
> Finally, the three connected subgraphs are randomly linked, resulting in a synthetic dataset that achieves local complexity through diverse generation methods (i.e., real-world topology, attribute similarity, and random connections) while enforcing global connectivity to maintain consistency. Additionally, this method retains original nodal features and labels, enhancing relevance for downstream tasks.
>
> We conduct additional experiments on this synthetic dataset under a semi-supervised node classification setting. As shown in the table below, at a 99.5% feature missing rate, FSD-CAP achieves an accuracy of 40.25%, significantly outperforming other diffusion-based methods such as FP (34.75%) and PCFI (35.25%). The results demonstrate that our progressive subgraph expansion mechanism remains effective in complex, mixed-structure environments, confirming the robustness and generalizability of Theorem 2 beyond standard benchmarks.
>
> **Table 1:** Accuracy of FP, PCFI and FSD-CAP on synthetic dataset for node classification task at 99.5% missing rate.
> | | FP| PCFI| FSD-CAP|
> | ----------------- | ------------- | ------------- | ------------- |
> | **synthetic dataset** | 34.75% ± 3.29 | 35.25% ± 3.07 | 40.25% ± 3.20 |
>
> **W2:** Thank you for your suggestion. We have added the algorithm pseudo-code for FSD-CAP to the appendix of the revised paper to more clearly illustrate its overall workflow.

---

> ### Author Response · Authors · 2025-11-23
> **Response to Reviewer Wp9X --- PartⅡ**
>
> **Questions:**
>
> **Q1:** Thank you for the comment. Whether a diffusion process operates in a super-diffusion regime is determined by how transition weights are assigned between nodes during message propagation. Traditional diffusion models such as Feature Propagation (FP) use the symmetric normalized adjacency matrix as the transition matrix, where each node distributes its information uniformly to its neighbors. This uniform averaging does not emphasize structurally important connections and tends to blur feature distinctions over iterations, which is not a super-diffusion regime.
>
> In contrast, we consider a fractional diffusion operator parameterized by $\gamma>0$. By applying a nonlinear transformation to the base diffusion weights, it amplifies differences among neighbors, enabling high-degree nodes to selectively aggregate more discriminative signals from low-degree neighbors. This shifts the propagation from uniform mixing to a strong-edge-dominated process, i.e., a super-diffusion regime, which helps alleviate over-smoothing.
>
> Notably, even with uniform diffusion, the class-refinement mechanism can still provide further performance gains by enhancing feature separation at the class level. To illustrate this, we conducted additional experiments on the baseline FP model. Under a semi-supervised node classification setting with 99.5\% of node features structurally missing, we compared the original FP with FP-2, an enhanced variant that incorporates our proposed class-level refinement mechanism, on five benchmark datasets. As shown in the table below, augmenting FP with class-level refinement (FP-2) yields noticeable improvements, confirming the complementary value of such mechanisms.
>
> **Table 1:** Accuracy(%) of FP and FP-2 on benchmarks for node classification task at 99.5% missing rate.
> | Dataset| FP| FP-2|
> |-------------|----------------|-----------------|
> | Cora| 72.71 ± 2.49| 79.04 ± 2.01|
> | CiteSeer| 57.98 ± 2.93| 69.89 ± 2.23|
> | PubMed| 74.18 ± 3.15| 74.78 ± 2.16|
> | Photo| 86.34 ± 1.09| 87.32 ± 1.35|
> | Computers| 77.19 ± 2.16   | 77.81 ± 1.65|
>
> **Q2:** Thank you for your suggestion. We have added the algorithm pseudo-code for FSD-CAP to the appendix of the revised paper to more clearly illustrate its overall workflow.
>
> **Q3:** Thank you for your thoughtful observation regarding the notation in Equations (3) and (4). Herein, we use lowercase $x_i^{(m)}(t)$ to denote the feature value of a specific node $v_i$ at iteration t within the m-th layer subgraph. This node-level formulation is intentionally adopted to clearly illustrate the fractional diffusion mechanism, namely how each node’s feature is updated based on its neighbors, observed entries, and information propagated from the previous layer.
>
> To avoid potential confusion and provide a more holistic view, we have now included a complete algorithmic description in the appendix (Algorithm 1), which presents the FSD-CAP update process in matrix form:
>
> $X^{(m)}(t) =  X^{(m)}(0) \odot M + \left(A^{\gamma,m}X^{(m)}(t-1) + \lambda X^{(m-1)}(K) \right) \odot (1 - M),  \quad t = 1, \dots, K$
>
> In the revised manuscript, we will explicitly distinguish between node-wise updates and batch-wise operations to improve readability.
>
> **Q4:** The referee's understanding is absolutely correct and sorry for the confusion. In FSD-CAP, the feature update for nodes in the $m$-th layer subgraph $G^{(m)}$ proceeds in two sequential steps during each iteration. Equation (3) performs fractional diffusion to obtain an intermediate feature estimate. Equation (4) then fuses this intermediate result with the converged features from the previous layer $(m-1)$ to enhance stability, while preserving observed entries via the mask. These two steps together constitute one complete iteration, and the procedure is repeated for $K$ iterations to obtain the converged output for the current layer. Thus, Equations (3) and (4) are applied in sequence, not as alternative or parallel definitions.
>
> To eliminate this ambiguity, we have revised the manuscript by unifying the two steps into a single expression:
>
>  $X^{(m)}(t) =  X^{(m)}(0) \odot M + \left(A^{\gamma,m}X^{(m)}(t-1) + \lambda X^{(m-1)}(K) \right) \odot (1 - M),  \quad t = 1, \dots, K$
>
> Additionally, we will include a detailed pseudocode in the appendix to clearly delineate the iterative procedure.

---

> > ### Comment · Reviewer_Wp9X · 2025-11-28
> >
> > Thank you to the authors for their response, the additional details and experiments have addressed my concerns about clarity and difficulty.
> >
> > However, I am unable to find any pseudocode in the currently available paper submission. Section 2 within the main paper is sufficient. However, algorithmic pseudocode will be helpful to aid independent technical implementations and results reproductions. This is the most critical concern for my current assessment of the paper.

---

> > > ### Author Response · Authors · 2025-11-28
> > >
> > > We sincerely appreciate your meticulous review and invaluable suggestions on our manuscript. We deeply apologize for the oversight of not including the pseudocode of the FSD-CAP algorithm in the appendix, as you pointed out.
> > >
> > > In response to your feedback, the complete pseudocode of the FSD-CAP algorithm has now been added to the latest revised appendix (A.6 and A.7), along with refined annotations and symbol explanations to enhance clarity and reproducibility.
> > >
> > > Once again, we extend our heartfelt gratitude for your expert guidance. Your insights have significantly improved the rigor and readability of our work. Should you have any further questions or require additional clarifications regarding the supplementary content, we remain fully committed to addressing them promptly.

---

### Author Response · Authors · 2025-11-28
**Kindly Invite Your Feedback**

Dear Reviewers,

We sincerely appreciate your thoughtful and constructive feedback. We have carefully addressed all concerns in our point-by-point responses.

As the end of the author-reviewer discussion phase is approaching, we kindly hope you have had a chance to review our updates. If there are any remaining questions or concerns, we are happy to discuss further and provide additional clarification.

Best regards,

Authors of ICLR 2026 Submission 12887

---

### Author Response · Authors · 2025-12-03
**General Response to Reviewers and Area Chair**

Dear Reviewers, AC, SAC, and PC,

We thank the reviewers for the thoughtful feedback. We study feature completion under extreme missingness and present three components: fractional-order diffusion, progressive subgraph expansion, and class-aware refinement. Each admits closed-form updates with convergence guarantees in Propositions 1–3 and Theorems 1–3. Taken together, these components provide a principled mechanism for amplifying long-range signals while avoiding over-smoothing.

FSD-CAP is a one-time preprocessing step: feature completion for a graph (Cora → Computers) averages 0.71s, after which the completed features can be reused by any downstream model. With only 0.5% observed features, FSD-CAP beats PCFI by 3.7% (structural mask) and 2.2% (uniform mask) on homophilic datasets, and by up to 8.6% on heterophilic graphs. As the missing rate rises from 60% to 99.5%, PCFI’s accuracy falls by 4.2% / 1.8% (structural / uniform), whereas FSD-CAP drops by just 1.5% / 0.5%, closing the gap to the full-feature GCN baseline from 4.9% to 1.3%. An ablation without the expansion step already surpasses PCFI, with all components active the full gain is +3.7%.

In the rebuttal, we added experiments and clarified the theory to address three main concerns: i) performance relative to recent state-of-the-art methods, ii) motivation and further validation of the first stage of FSD-CAP, and iii) generalization to heterophilous graphs.
- **Performance relative to recent SOTA methods (Reviewer 66Pn, rfQB):** Beyond the baselines reported in the manuscript, we additionally compare FSD-CAP with latest methods, FISF [1] and GOODIE [2], both published in 2025, on node classification and link prediction. As detailed in responses (Part II) to reviewer 66Pn, FSD-CAP consistently outperforms these strong competitors across structural and uniform missing settings, demonstrating its effectiveness in highly incomplete regimes. Below, we summarize the average performance of FISF, GOODIE, and FSD-CAP over 10 runs on node classification and link prediction across the datasets (Cora, CiteSeer, PubMed, Photo, and Computers) under structural and uniform missingness.

Table 1. Average accuracy (%) comparison at 99.5% missing rate for node classification task.
||FISF|GOODIE|FSD-CAP|
|--------------------|-----|------|---------|
|Average (structural)|78.27|74.67|**80.06**|
|Average (uniform)|79.12|77.06|**81.01**|

Table 2. Average AUC score (%) comparison at 99.5% missing rate for link prediction task.
||FISF|GOODIE|FSD-CAP|
|--------------------|-----|------|---------|
|Average (structural)|89.03|72.75|**91.65**|
|Average (uniform)|89.60|78.05|**92.41**|

- **Motivation and further validation of the first stage (Reviewer Wp9X, 66Pn):** As in our response to Reviewer 66Pn (Part IV), the fractional subgraph diffusion is motivated by epidemic dynamics and modulates propagation sharpness by redefining how weights are allocated, thereby mitigating over-smoothing. To further validate its behavior under complex local and global structure, we construct a synthetic graph from PubMed that mixes real, similarity-based, and random subgraphs. As shown in our response to Reviewer Wp9X, the progressive expansion remains effective in this mixed-topology setting, demonstrating robustness beyond standard homophilic benchmarks.

- **Generalization to heterophilous graphs (Reviewer 66Pn, vLJV):** We evaluate FSD-CAP on four standard heterophilous benchmarks. As shown in our detailed replies, FSD-CAP consistently surpasses not only baselines (FP, PCFI) and heterophily-oriented GNNs (Hop-GNN, GPR-GNN), but also recent SOTA methods like FISF and GOODIE. Averaged across the four datasets, FISF obtains 50.51\% / 49.43\% (structural / uniform), which is 5.71 / 9.42 percentage points below FSD-CAP. For GOODIE, the gap even widens to 10.13 / 12.87 percentage points. These findings are consistent with our design. Progressive subgraph diffusion limits early cross-class leakage, and fractional diffusion allows milder mixing on weak homophily. The class-aware refinement then pulls uncertain nodes toward class prototypes and downweights boundary nodes by entropy.

Our contribution goes beyond empirical gains. We introduce fractional diffusion to tune propagation sharpness and a class-aware refinement step that improves feature discriminability, and we analyze both theoretically. Experiments on homophilic, heterophilic, and synthetic graphs, together with feature-space visualizations, show that the method remains effective even under extreme sparsity.

Again, we thank the referee for their diligence and the opportunity to strengthen the paper.

Best regards,

the authors

**References:**

[1] Um, Daeho, et al. "Propagate and Inject: Revisiting Propagation-Based Feature Imputation for Graphs with Partially Observed Features." ICML 2025.

[2] Yun, Sukwon, et al. "Oldie but Goodie: Re-illuminating Label Propagation on Graphs with Partially Observed Features." KDD 2025.

---

### Meta-Review · Area_Chair_CP3w · 2026-01-08

**Summary:**

This paper addresses the challenge of graph feature imputation under extreme missing rates. The authors propose FSD-CAP, a two-stage diffusion-based framework for imputing missing node features, especially in the cases with high sparsity of node metadata, by applying 1) Factional diffusion operator, 2) Progressive subgraph diffusion, and 3) Class-level feature refinement. Experiments demonstrate the effectiveness of FSD-CAP on semi-supervised node classification and link prediction at a high missing rate.

**Reviewer Concerns:**

The concerns are around performance relative to recent state-of-the-art methods, motivation and further validation of the first stage of FSD-CAP, generalization to heterophilous graphs, label availability and three conditions, empirical completeness and presentation. The authors have tried to address most concerns and strengthened the paper by providing additional experimental results requested by the reviewers and tried to argue with the reviewers. However, some concerns as follows seem still unlearly addressed to me:

[Reviewer 66Pn] "its performance is likely to depend on label availability". The authors responded "we followed the semi-supervised setting adopted in prior work" and provided additional results under varying feature missing rates and label rates, but didn't directly anwser the reviewer's concern on what if no labels are available.

[Reviewer vLJV] "the methods succeed only when three conditions hold: (1) the node features are informative for the task at hand; (2) the features are homophilic (adjacent nodes tend to share similar values); and (3) the graph exhibits community structure in which each community is dominated by nodes with the same feature label". The authors responded "our approach implicitly assumes a degree of feature homophily, that is, connected nodes tend to have similar attributes" and ackknowledged "As the reviewer points out, when these assumptions are violated, particularly in heterophilic graphs, where neighboring nodes often belong to different classes, the effectiveness of smoothness-based diffusion may degrade". The response implies that the authors acknowledged the limitation of the proposed method.

**Reviewer Scores:**

Reviewer Wp9X (score 6 confidence 4) is likely to keep the score unchanged as he/she slightly lean to accept after post-rebuttal discussion.

Reviewer 66Pn (score 2 confidence 4) might slightly raise the score to 4 as the authors have responded to most his/her concerns with additional results.

It is hard to predict Reviewer rfQB's score (score 4 confidence 4) becuase he/she raised a lot of concerns.

Reviewer vLJV (score 4 confidence 4) might slightly raise the score to 6 as he/she responded that the authors had addressed all his/her concerns.

Reviewer fH1t (score 4 confidence 1) is likely to raise the score to 6 as his/her confidence is very low.

---

### Decision · Program_Chairs · 2026-01-26

Accept (Poster)